# Cloud liquid water path in the sub-Arctic region of Europe as derived from ground-based and space-borne remote observations

Vladimir S. Kostsov[1], Anke Kniffka[2], Dmitry V. Ionov[1]

[1]Department of Atmospheric Physics, St. Petersburg State University, St.Petersburg, 198504, Russia

[2]Institute of Meteorology and Climate Research, Karlsruhe Institute of Technology, Karlsruhe, 76131, Germany

*Correspondence to: Vladimir S. Kostsov (vlad@troll.phys.spbu.ru) and Anke Kniffka (anke.kniffka@kit.edu)*

**Abstract.** Tropospheric clouds are a very important component of the climate system and of the hydrological cycle in the Arctic and sub-Arctic. Liquid water path (LWP) is one of the key parameters of clouds urgently needed for a variety of studies, including the snow cover and climate modelling at Northern latitudes. A joint analysis of the LWP values obtained from observations by the SEVIRI satellite instrument and from ground-based observations by the RPG-HATPRO microwave radiometer near St.Petersburg, Russia (60N, 30E) has been made. The time period of selected datasets spans two years (December 2012 – November 2014) excluding winter months, since the specific requirements to SEVIRI observations restrict measurements at Northern latitudes in winter when the solar zenith angle is too large. The radiometer measurement site is located very close to the shore of the Gulf of Finland, and our study has revealed considerable differences between the LWP values obtained by SEVIRI over land and over water areas in the region under investigation. Therefore, special attention has been paid to the analysis of the LWP spatial distributions derived from SEVIRI observations at scales from 15 km to 150 km in the vicinity of St.Petersburg. A good agreement between the daily median LWP values obtained from the SEVIRI and the RPG-HATPRO observations has been shown: the RMS difference has been estimated as 0.016 kg m$^{-2}$ for a warm season and 0.048 kg m$^{-2}$ for a cold season. During seven months (February – May and August – October), the SEVIRI and the RPG-HATPRO instruments revealed similar diurnal variations of LWP, while considerable discrepancies between the diurnal variations obtained by the two instruments have been detected in June and July. On the basis of reanalysis data, it has been shown that the LWP diurnal cycles are characterized by a considerable interannual variability.

## 1 Introduction

The interest of studying the sub-Arctic atmosphere is enhanced due to the so-called "arctic amplification" effect. This feedback effect is expected to enhance the response of the arctic climate system to both anthropogenic and natural forcing compared to the planet as a whole. The large seasonal and interannual variation in low- and high-pressure systems and associated environmental variability due the location of the Baltic Sea between the North Atlantic and Eurasian air masses makes Northern Europe especially important to study atmospheric processes (Eriksson et al, 2007). Callaghan et al. (2010) applied statistical techniques to the long-term data obtained in the Swedish sub-Arctic and noted that changes in climate were associated with reduced temperature variability, particularly the loss of cold winters and cool summers, and an increase in extreme precipitation events that cause mountain slope instability and infrastructure failure. The findings of Callaghan et al. (2010) have demonstrated that the recent warming period is very different in character from that in the late 1930s and early 1940's and that we could be now entering a new climate era.

In the studies devoted to the possible consequences of climate change, much attention is paid to the hydrological cycle due to the fact that the snow cover influences greatly surface albedo, vegetation period, soil warming/freezing and ecosystems. Dankers and Christensen (2005) presented a model-based assessment of the impact of climate changes on sub-Arctic hydrology in the separate region of Northern Finland and Norway. The impact of climate change on snow cover and soil temperatures in the high latitude regions has been studied by Mellander et al. (2007) for different scenarios of $CO_2$ emission.

Clouds are a very important component both of the climate system and the hydrological cycle since interactions between clouds and seasonal snow cover are expected to have a significant effect on climate and its variation in the Arctic and sub-Arctic (Zhang et al., 1996). On the basis of model calculations Zhang et al. (1996) have shown that the melting rate of the snowpack increases for thin clouds and decreases for thick clouds with increasing liquid water path (LWP). Moreover, clouds may have a negative impact on snowmelt when LWP is very large. Beesley (2000) has presented the results of model studies of the effect of clouds on the ice thickness in Arctic and has shown, in particular, that interactions between the Arctic and midlatitudes are considerable and very important.

The clouds in Arctic and in sub-Arctic are intensively studied using different observation platforms and remote sensing methods. Garrett and Zhao (2013) described a method of retrieving various parameters of thin clouds from ground-based infrared radiation measurements by an interferometer and applied this method to observational data obtained in Alaska (the limitation of the method is the inapplicability to thick clouds that radiate as a blackbody). The cloud liquid water path is one of the target atmospheric parameters obtained from microwave (MW) remote measurements. Several ground-based microwave radiometers are permanently functioning at Northern latitudes as the elements of the MWRnet - An International Network of Ground-based Microwave Radiometers (http://cetemps.aquila.infn.it/mwrnet/main_files/MWRnetmap.html), however the measurement network is rather coarse in that region. Special measurements campaigns with microwave radiometers have been carried out in Europe with the focus on liquid water path and the difficulties have been demonstrated to compare the measured LWP with models of different grid size (Meijgaard and Crewell, 2005). Microwave radiometers delivering information on LWP are functioning also on board satellites. Besides, there are satellite instruments that measure LWP using other electromagnetic ranges (visible-infrared). The climatology of LWP obtained from satellite observations have been presented by Elsaesser et al. (2017). The importance of combining visible-infrared imager data and passive microwave LWP observations for estimating uncertainties and improving the accuracy of these observations has been demonstrated by Greenwald et al., 2018.

It should be emphasized that LWP is an essential climate variable and the assessment and improvement of the accuracy of LWP data obtained from different platforms and instruments is still an actual problem. Lohmann and Neubauer (2018) have reported that global annual mean LWP values over oceans derived from measurements by different satellite sensors have very broad range of $30 - 90$ g m$^{-2}$; besides, both retrievals from visible–near-infrared sensors and microwave sensors have biases in LWP data. The validation campaigns for LWP measurements from space often use ground-based LWP observations by microwave radiometers as the reference data since they have a precision that is superior to current satellite remote sensing techniques (Roebeling et al., 2008a).

Along with the high accuracy of LWP retrievals, other advantages of the ground-based MW observations should be mentioned. Ground-based MW instruments operate with very high temporal resolution (1-2 second), continuously for very long periods of time, in unattended mode, independently of solar illumination and nearly at all weather conditions. The evident advantage of satellite observations is their global scale, however the MW satellite sensors deliver the information only over water areas since the emissivity of the land surface is highly variable. The superiority of the SEVIRI instrument working in visible–near-infrared range is the possibility to make observations over water areas and land surface as well, however only when the atmosphere is illuminated by Sun since the instrument measures the reflected solar radiation.

This study exploits the LWP measurements made by the ground-based microwave radiometer RPG-HATPRO (Radiometer Physics GmbH - Humidity And Temperature PROfiler) operating at the measurement site of St.Petersburg State University, Russia, and the satellite measurements made by the SEVIRI (Spinning Enhanced Visible and InfraRed Imager) instrument over the area in the vicinity of St.Petersburg (60$^{o}$N, 30$^{o}$E). The geographical area under investigation can be considered as belonging to sub-Arctic region of Europe if we use the general definition of sub-Arctic region as a territory located in the

latitude range $50^{o}$-$70^{o}$. The comparisons of the LWP measurements made in the present study are to a certain extent the continuation of the series of investigations that have been done previously by Roebeling et al. (2008a, 2008b), Greuell and Roebeling (2009). Roebeling et al. (2008a) determined the accuracy and precision of LWP retrievals from SEVIRI on board Meteosat-8 using 1 year of LWP retrievals from microwave radiometer measurements of two CloudNET (http://www.cloud-net.org/) stations located in the United Kingdom (Chilbolton) and France (Palaiseau). The obtained results have been generalized as related to Northern Europe. In particular, the overestimation of instantaneous LWP values by SEVIRI was detected during winter, and this overestimation has been suggested to be caused by neglecting cloud inhomogeneities in the SEVIRI retrieval algorithm. It should be emphasized that the microwave ground-based measurements were selected as a reference for validation since this type of measurements has a precision that is superior to current satellite remote sensing techniques (Roebeling et al., 2008a). Roebeling et al. (2008b) examined the consistency between LWP and geometrical thickness values inferred from the SEVIRI measurements. In the study by Roebeling et al. (2008b), the LWP and geometrical thickness from the SEVIRI retrievals were compared to a statistically significant set of collocated and synchronized ground-based measurements at two above mentioned CloudNET stations. The dual-channel passive microwave radiometers of Chilbolton (22.2 and 28.8-GHz) and Palaiseau (24 and 37-GHz) were used for the ground-based observation of LWP while geometrical thickness was obtained from cloud radar and lidar observations. Greuell and Roebeling (2009) investigated in great detail the important problem of working out the standards for validation of the LWP measurements by the SEVIRI instrument. Since the differences between the satellite-derived and the ground-based LWP values are partly associated with the validation procedure itself through the scale difference and parallax effect, minimizing these types of differences is the essential part of any comparison. Greuell and Roebeling (2009) have proposed to perform averaging of the ground-based microwave measurements with a Gaussian weight function, by using a time scale that is considerably longer than the time of the cloud movement across the validation area (by a factor of 3–15).

Similar to the articles by Roebeling et al. (2008a, 2008b), and Greuell and Roebeling (2009), the present article is focused on the comparison of the LWP values obtained by the SEVIRI satellite instrument and the ground-based microwave radiometer. However, there are two important peculiarities:

1) The latitude of the St.Petersburg measurement site is higher than the latitudes of Chilbolton and Palaiseau stations. As a result, the solar zenith angle values are larger, that can lead to the increase of the SEVIRI measurement errors. Also the size of the footprint is larger than at the other locations. The validation of the SEVIRI measurements under these conditions is important for the problem of monitoring the sub-Arctic territories.

2) The St.Petersburg measurement site is located at about 3 km distance from the shore of the Gulf of Finland while the above mentioned stations are located far from large water areas. Since the LWP values can be essentially different over land and sea surfaces (Karlsson, 2003), and taking into account the finite spatial resolution of the satellite observations, one can suggest that the validation procedure becomes more difficult under such conditions. In the present study, much attention is paid to the investigation of this problem.

So, the main goals of the present study were to identify the problems of the comparison of HATPRO and SEVIRI measurements of LWP at high latitudes over the complex terrain which includes land and water areas, to analyse the frequency distributions and diurnal cycles derived from measurements of the two instruments and to assess systematic and unsystematic discrepancies between the satellite and ground-based data sets.

## 2 Dataset description

### 2.1 RPG-HATPRO original data

The 14-channel microwave radiometer RPG-HATPRO (generation 3) developed for the retrieval of temperature and humidity profiles in the troposphere along with LWP and integrated water vapour (Rose et al., 2005) has been routinely functioning at the measurement site of St.Petersburg State University (59.88$^{o}$N, 29.83$^{o}$E) since June 2012 with a sampling interval about 1-2 s and an integration time 1 s. The complete description of radiometers of the HATPRO type (below we shall omit "RPG-" for simplicity) can be found at the web-site of the manufacturer (http://www.radiometer-physics.de). All information on experimental setup and the measurement site can be found in the paper by Kostsov et al. (2016). The LWP values are derived from the microwave radiation brightness temperature measurements by two separate and independent retrieval algorithms. The first algorithm is the built-in regression technique that had been developed by the manufacturer. The second algorithm is based on the inversion of the radiative transfer equation (so-called "physical algorithm") and uses the well known and widely applied approach of simultaneous retrieval of profiles of several atmospheric parameters that influence the radiative transfer at frequencies corresponding to spectral channels of a radiometer. The description of the physical algorithm, estimation of the retrieval accuracy for different parameters and the examples of retrievals can be found in the paper by Kostsov (2015a). The multiparameter retrieval procedure accounting for the a priori information of different types that had been used in the physical algorithm is presented in detail in the paper (Kostsov, 2015b). The results of the cross-validation of the regression algorithm and physical algorithm are described in the article by Kostsov et al. (2018). Kostsov et al. (2018) have found out that the response of the regression algorithm to artefacts in the input data is considerably larger than the response of the physical algorithm. Also, there are problems with the detection of cloud-free periods from the data obtained by the regression algorithm. The conclusion has been made that the utilization of the physical algorithm is more preferable. Therefore, only the results obtained by the physical algorithm have been used in the present study.

The estimations of the accuracy of LWP retrievals by the HATPRO radiometer near St.Petersburg have been made previously (Kostsov et al., 2017) on the basis of the analysis of cloud-free situations and on the basis of calculations of the error matrix of the physical algorithm. The analysis of cloud-free situations has shown 0.009-0.011 kg m$^{-2}$ for bias and 0.001 kg m$^{-2}$ for random error. It should be noted that the corresponding values reported in the study by Matzler and Morland (2009) are 0.002-0.005 kg m$^{-2}$ and 0.001 kg m$^{-2}$. The error matrix calculations have shown that the random error varies in the range 0.001 – 0.008 kg m$^{-2}$ for all observed LWP values (up to 1 kg m$^{-2}$). Cossu et al. (2015) obtained the slightly higher bias of LWP retrievals by ground-based MW radiometry which constituted 0.01-0.02 kg m$^{-2}$ and they also estimated the random error as 10-20% for LWP greater than 0.1 kg m$^{-2}$.

The time period 1 December 2012 – 30 November 2014 is considered in the present study due to the following reasons: (1) the instrument was functioning without failures and interrupts, (2) the obtained data volume is sufficient for derivation of statistical characteristics, (3) the measurement data have already been validated and analysed for this time period (Kostsov et al., 2018). There were 13 calibrations of the instrument during this period of time including 7 absolute calibrations with liquid nitrogen and 6 sky-tipping calibrations. The interval between absolute calibrations varied from 2 to 4 months.

### 2.2 SEVIRI original data

The SEVIRI-derived LWP measurements are part of the climate data record CLAAS 2 (CLoud property dAtAset using SEVIRI – Edition 2). It was created by the Satellite Application Facility on Climate Monitoring (CM SAF) based on the SEVIRI measurements on the geostationary MSG satellites (Benas et al., 2017 and Stengel et al., 2014). SEVIRI scans the earth in 12 spectral channels, ranging from the visible (0.6 μm) to the near infrared (14.4 μm) in the electromagnetic

spectrum with bandwidths between 0.14 and 2 µm. The scans are taken with a temporal resolution of 15 minutes. The ground pixel size varies from 3 km at nadir to about 11 km near the edge of SEVIRI's field of view. In the vicinity of St.Petersburg the ground pixel size is about 7 km. CLAAS data record was created from measurements of all SEVIRI instruments onboard the MSG 1-3 satellites and covers the time-span 2004 – 2015. It was processed using a single retrieval system with an instrument intercalibration based on MODIS Aqua (Meirink et al., 2013) data to ensure the exclusion of artificial temporal inhomogeneity.

The Cloud Physical Properties (CPP) retrieval algorithm uses the channels at 1.6 µm and 0.6 µm. In the visible channel mainly the influence of the cloud's optical thickness is translated into reflectance, whereas in the near infrared the variation of reflectance is caused by variations in effective radius of the clouds. With the help of detailed radiative transfer calculations, look-up tables were created and the observed reflectances are interpolated in between. The LWP data obtained by SEVIRI have already been used in a number of studies of the temporal and spatial characteristics of clouds of different types, in particular by Kniffka et al. (2014).

In the validation document of CM SAF (Finkensieper et al. 2016), the bias of the LWP measurements is specified to be 0.00007 kg/m² for monthly mean values compared to MODIS and the bias-corrected root mean square error amounts to 0.0101 kg/m². Here the complete field of view of SEVIRI and the monthly means from 2004 – 2015 were analysed. A comparison with AMSR-E was also conducted and showed a bias of 0.0034 kg/m² and a bias-corrected root mean square error of 0.034 kg/m². Unfortunately, this comparison was based only on a single overpass of AMSR-E. In Roebeling et al. (2008a) comparisons were made for the three sites Cabauw (Netherlands), Chilbolton (United Kingdom) and Palaiseau (France) for time-series of 4 years. Here the bias was found to be 0.005 kg/m² in summer and 0.010 kg/m² in winter while the variance was stable with 0.030 kg/m². Please note that the latter study was based on a retrieval algorithm state 10 years ago, until today, the retrieval has undergone many modifications that led to an overall improvement.

In the present study, non-averaged LWP and CPH (cloud phase) fields (level 2 data) from the CLAAS 2 dataset were used for the time period of ground-based original data (1 December 2012 – 30 November 2014).

**2.3 Data selection procedure and datasets for comparisons**

The high quality of ground-based MW measurements has been taken as a main criterion used in the data selection procedure. This criterion included the fulfilment of three requirements:

1)  The measurement days must have been completely rain free. It means that all rain flag values must have been equal to zero from 00:00:00 UTC till 23:59:59 UTC for every specific day. The reason to completely exclude from consideration the days with rains is the following. Not only during a rain event but also for a rather long period of time after it, the data provided by HATPRO are erroneous since the radio dome of the instrument is wet. The duration of this after-rain period for St.Petersburg site has been estimated in the study by Kostsov et al. (2017) as 4-6 hours. Moreover, it has been demonstrated in the mentioned study that the situations are possible, when the measurements are erroneous even before the rain event, when the rain sensor is not yet detecting the rain signal. It is evident that even one rain event during a day results in the considerable loss of data. It should be mentioned that SEVIRI retrievals fail in cases of strong vertical LWP gradients and especially during rain events.

2)  The measurement process must not have had gaps which are defined as 15 min or more period without measurements.

3)  The quality flag of MW measurements must have been zero for all retrievals that means the successful convergence of the iteration process of physical retrieval for every single measurement.

The first and second requirements are important since the MW measurements should be averaged over the time period of several dozens of minutes in order to be consistent with a single pixel measurement made from space. Rain events and gaps

in measurement process can spoil the results of averaging. Also, for the estimation of the mean diurnal cycle, it is desirable that all days have uninterruptible flow of measurements. The absence of rains and measurement gaps ensure meeting such condition.

The specific requirements to SEVIRI observations restrict measurements just after sunrise and before sunset when the solar zenith angle (SZA) is too large. Therefore, all MW and satellite measurements when SZA was greater than 72 degrees have been excluded from consideration as it was done in the study by Roebeling et al. (2008a). As a result, no measurements during winter months December and January could be selected for analysis, and the number of measurements selected in February was small.

The sampling interval (the interval between instantaneous measurements) of routinely performed ground-based MW observations is about 1-2 s since the sampling period (the integration time of the incoming atmospheric signal) is equal to 1 s. It is important because the retrieval algorithms are typically non-linear and therefore the retrieval needs to be made on high temporal resolution brightness temperature data for subsequent averaging of the results but not vice versa. Also, it should be mentioned that situations can be very different and in particular convective boundary layer clouds have high variability. However, there are situations when keeping the sampling interval as small as possible is problematic. If an instrument is functioning in the mode of azimuth scanning or zenith scanning, some time is needed to change pointing. Also, an instrument can be set to make the mixed mode observations. In this case the interval between measurements made in a certain mode can be rather large. It has been noted by Rose et al. (2005) that the integration time (and also the sampling interval) should not be greater than 20 s in order that the short-period variations of tropospheric humidity and cloud liquid water can be registered, and in this case the temporal resolution is comparable to the resolution of state-of-the-art numerical weather prediction (NWP) models. Kostsov et al. (2016) applied the information theory approach to the analysis of the ground-based MW measurements and performed calculations of the information volume for datasets with different sampling intervals. The obtained results have shown that even for constant atmospheric conditions the sampling interval should not be greater than 100-200 s in order that maximum information could be extracted from MW measurements. Though this conclusion had more theoretical value than a practical one, for the present study we have chosen two original MW datasets that differ by the sampling interval: 120 s and 10 s.

There are several slightly different schemes for averaging MW data in order that the resulting value best represent the LWP obtained by the SEVIRI instrument for one pixel. Roebeling et al. (2008a) reported that averaging the MW retrievals of LWP had been done over 20 min period assuming the wind speed about 10 m s$^{-1}$ and the SEVIRI field of view (4 x 7 km$^2$). In the study (Roebeling et al. 2008b), the time period of 30 min has been mentioned as the period taken for averaging. Greuell and Roebeling (2009) proposed to compute the ground-based LWP by averaging the MW measurements with a Gaussian weight function, by using a time scale that is considerably longer than the time during which the clouds move across the validation area (by a factor of 3–15). Simultaneously, they recommended computing the satellite data by averaging the LWP retrieved by SEVIRI over the pixels surrounding the ground station by means of a Gaussian weight function with a length scale defining the validation area. The comprehensive discussion of the aggregation of the ground-based LWP data to coarser time scales has been presented by Meijgaard and Crewell (2005) who considered the comparison of ground-based LWP observations with the estimates from NWP models. Taking into account the mentioned findings, in the present study we used two schemes with different averaging periods of 20 min and 60 min, however the weighting function has been taken not Gaussian but a boxcar for simplicity. All data selection steps are summarised in Table 1, this table presents also the designation of four HATPRO datasets HAT$_{n-m}$ used for comparison. It should be noted that all comparisons have been made for SEVIRI ground pixel which is the nearest to the radiometer site. In case other pixels are considered, it will be mentioned explicitly.

Simultaneously with synchronisation between the HATPRO and SEVIRI values of LWP, control of the cloud phase has been made. The algorithm that is used for processing raw data obtained by the SEVIRI instrument delivers the parameter CPH which identifies the cloud phase at the cloud top. The CPH values 0, 1, and 2 correspond to clear case, liquid phase and ice crystals. Only liquid phase clouds have been considered, therefore all SEVIRI measurements with CPH=2 have been excluded from further analysis and from synchronization with HATPRO results.

Every $HAT_{n-m}$ dataset has been divided into two ensembles corresponding to different scenarios of observations (seasonal periods). The description of these periods is given in Table 2. The division has been done on the basis of atmospheric temperature and humidity criterion: the data have been attributed either to warm and humid (WH) or to cold and dry (CD) period. The corresponding time intervals are: 1 May – 30 November and 1 December – 30 April. The mean vertical distribution and standard deviation of temperature and humidity for the mentioned periods can be found in the paper by Kostsov et al. (2016). As one can see from Table 2, the number of selected days during the WH period is noticeably larger than during the CD period. The total number of 210 days means that about 28 % of the whole 2-year dataset is suitable for comparative analysis.

**3 Data overview. LWP differences over sea and land**

Fig. 1 shows the location of 441 SEVIRI measurement pixels selected for analysis of the large terrain surrounding St.Petersburg and the location of nine pixels corresponding to the small terrain in the vicinity of the radiometric measurement site. The large terrain comprises parts of the Gulf of Finland, Karelian Isthmus and Ladoga Lake and the region to the South and South-West of St.Petersburg. The small terrain size is about 20x20 $km^2$. The Northern part of it is a water area and the Southern part is a land area. The radiometer is located close to the shore of the Gulf of Finland at a distance of 2.7 km from the coastline. The centre of pixel 243 is the nearest to the measurement site, the distance is 1.5 km. The accuracy of SEVIRI's geolocation depends on the actual satellite on which the instrument is mounted and amounts approximately to 1.32 km in north-south direction and 0.15 km in east-west direction as stated in the document on MSG level 1.5 image data description (EUMETSAT, 2017) plus an additional error of 1.5 km in both directions in the data prior to 2017 because of an undetected pixel offset..Greuell and Roebeling (2009) studied the influence of the parallax effect (the horizontal displacement of a cloud viewed by a ground-based radiometer in a satellite image) on the results of the comparison of the data obtained by SEVIRI and ground-based microwave radiometers. Obviously, this influence is not significant for homogeneous cloud fields, and for clouds at low heights. The estimations of the parallax effect for Chilbolton and Palaiseau stations made by Greuell and Roebeling (2009) in terms of horizontal displacement were 3.1 km and 2.6 km correspondingly assuming a cloud top height of 3 km. Based on these values and accounting for the higher latitude of the St.Petersburg measurement site, we can expect the parallax effect for the St.Petersburg measurement site to be about 3 km or more in terms of the displacement to the North direction. This means that the satellite image point corresponding to a cloud view by the HATPRO radiometer is located over the coastline or over the water of the Gulf of Finland.

Before performing any comparisons of the satellite and ground-based data we analysed the spatial distribution of the LWP values obtained from the SEVIRI measurements over the large and small terrains as defined in Fig. 1. Fig. 2 presents the maps of the mean LWP values calculated for the large and small terrains and for the whole considered 2-year period of observations (about 20000 data points per pixel). Comparing Figs. 1 and 2, one can see that the difference between the LWP over land and over water is clearly visible. Over water areas, the mean LWP value is less than 0.075 kg $m^{-2}$, while the LWP exceeds these value over land. In order to assess whether this gradient can influence the results of the comparison of the SEVIRI and the HATPRO data, we plotted similar maps only for the data selected for comparison and considered the WH and the CD periods separately (about 4000 and 2000 data points per pixel respectively), see Fig. 3. The land-sea gradient of

the mean LWP values can be seen for both periods. The magnitude of the land-sea difference for mean LWP values (0.032 kg m$^{-2}$) is comparable to the value calculated for the whole 2-year period of observations (0.040 kg m$^{-2}$). However, the mean LWP values themselves are much lower than obtained for the whole 2-year period. This result is obvious since the rainy days have been excluded from analysis that means the presence of a large number of low-LWP and clear atmospheric conditions in the selected ensembles. If we consider the CD period, we can see that the land-sea gradient in the mean LWP values is noticeably lower than for the WH period. The similar maps of the median LWP values are given in Appendix 1.

It should be noted that the land-sea differences of cloud characteristics in Northern Europe have been detected earlier. Karlsson (2003) compiled regional cloud climatologies covering the Scandinavian region on the basis of processing data from the NOAA Advanced Very High Resolution Radiometer (AVHRR) instrument for the period 1991–2000. Considerable local-scale variation of cloud amounts was found in the region. During the spring and summer seasons, as a contrast to winter and autumn conditions, much less cloudiness has been found over seawater and major lakes. It has been suggested that the cold sea surface temperatures in the Baltic Sea (especially in spring and early summer due to inflow of cold fresh water from melting snow) lead to a considerable stabilization of near-surface layer of the troposphere. This explanation agrees well with the fact detected in our study: the land-sea gradient in the mean LWP values for the CD period. is noticeably lower than for the WH period.

Taking into account the estimated considerable values of the parallax effect and the land-sea LWP gradient, one can come to the conclusion that the combination of both in specific cases can influence the results of the comparison of the SEVIRI and the HATPRO data. In order to investigate this possible influence, not one, but two SEVIRI pixels have been chosen for analysis: 243 and 221. For simplicity, below we shall refer to the "main" pixel 243 as to pixel 0.

Concluding this section, we consider how different schemes of sampling and averaging of the HATPRO original data influence the HATPRO LWP values taken for comparison with the results obtained by SEVIRI. Two scatter plots are presented in Fig. 4. The first scatter plot shows the LWP values contained in datasets HAT$_{10-20}$ and HAT$_{120-20}$ (see Table 1) and gives the impression of the influence of the sampling interval on the data averaged over the same time period (in the considered case averaging over the 20 min period was done and the two sampling intervals were compared – 10 s and 120 s). One can see that this influence is noticeable but not as strong as the influence of the averaging period, which is illustrated by the second scatter plot. The second plot displays the LWP data sampled every 10 s but averaged over 20 min and 60 min respectively. The maximal difference between LWP values in this case can reach 50 % and more. Therefore, for comparisons of HATPRO and SEVIRI data, we took only two of four datasets that have the same sampling interval (10 s) but different averaging period (20 min and 60 min): HAT$_{10-20}$ and HAT$_{10-60}$.

**4 Case study**

Since LWP, spatial distribution, and temporal evolution of clouds are highly variable characteristics, the analysis of specific atmospheric conditions (case study) can be very useful for understanding how different factors influence the results of the comparison of the ground-based and satellite data.

First of all, in order to have the impression of the overall agreement of the HATPRO and the SEVIRI data during different seasons let us consider the daily median LWP values obtained by SEVIRI and HATPRO. These values are shown in Fig. 5 as a function of day sequence number, which corresponds to the simple consecutive enumeration of days in the datasets. Also, the figure presents the distribution of days in the datasets over months. This distribution is practically uniform in the WH dataset, but it should be noted that a relatively large number of measurements were suitable for comparison in July and in June only a few measurements were suitable for comparison. In the CD dataset, the measurements in December and

January are not present at all because of large SZA that restricts the SEVIRI observations. There are only a few measurements suitable for comparison in February but there are a large number of measurements in April and March. We note that March is one of the most cloud-free months in St.Petersburg but according to selection criteria (see Section 2) the cloud-free days have been included in the datasets also. As far as mostly clear-sky conditions are concerned (median LWP close to zero), we note that the agreement of HATPRO and SEVIRI data for these situations is very good, that can be seen from Fig. 5. This conclusion is valid for both seasons. For cloudy days, the agreement is noticeably better during the WH season excluding day No 52 when the difference between the SEVIRI and the HATPRO LWP values is very large and constitutes about 0.4 kg m$^{-2}$.

The estimates of the bias and rms difference between the daily median LWP values derived from satellite and ground based observations are given in Table 3. Since there was only one day with extremely large discrepancy between the results (day No 52), we excluded this day from the calculations (this is 14 May 2014 and the reasons for large discrepancies are discussed below). The daily median values averaged over the datasets constitute 0.017 kg m$^{-2}$ and 0.02 kg m$^{-2}$ for WH and CD datasets correspondingly. The values of the difference calculated for the HAT$_{10-20}$ and the HAT$_{10-60}$ datasets are very close, so the preference can be given neither to the averaging of the radiometer data over 20 min interval nor to averaging over 60 min interval. The RMS difference has been estimated as 0.016 kg m$^{-2}$ for a warm season that is considerably lower than the RMS difference for a cold season which is 0.048 kg m$^{-2}$. The bias is very small and it is negative for the WH season and positive for the CD season. It should be emphasized that the correlation coefficients for the WH season are considerably larger than for the CD season.

Fig. 6 presents the examples of instantaneous measurements of LWP by SEVIRI and by HATPRO (two HATPRO datasets were used: HAT$_{10-20}$ and HAT$_{10-60}$) for several days of the WH season. These days have been selected for the purpose of demonstrating the cases with good and bad agreement between the data. A very good agreement can be seen on 6 May 2013 and on 6 June 2013. On 6 May 2013 the clouds were present in the early morning and the rest of the day was cloud-free. On 6 June 2013 the clouds appeared in the afternoon and disappeared in the evening. For both cases the qualitative and quantitative agreement of the HATPRO and the SEVIRI results can be considered as excellent. It should be taken into account that the day fraction is bound to the UTC, not the local time, the time difference is 3 hours. A very good agreement is also demonstrated on 11 October 2014 when HATPRO and SEVIRI show two maxima of LWP during the day, however the second maximum is narrower for HATPRO. The day 5 October 2014 presents an example of the combination of good and bad agreement between the data. Most of the time, HATPRO and SEVIRI show the same very smooth temporal behaviour of LWP, but in the late afternoon sudden oscillations appear in the SEVIRI data.

The examples of the considerable disagreement between the HATPRO and the SEVIRI data are the measurements during two days: 14 May 2014 and 2 July 2014. On 14 May 2014 the LWP was nearly constant and close to 0.25 kg m$^{-2}$ according to the HATPRO radiometer observations while SEVIRI provided much higher quantities (with the peak of 2 kg m$^{-2}$) most of the time except 3 hours in the evening. On 2 July 2014 both instruments detected high variability of LWP with the same magnitude, but there was no correlation between the satellite and the ground-based measurements. In order to identify the reasons of strong discrepancies between the data we have analysed the results of meteorological observations at the St.Petersburg meteorological station (WMO ID 26063) during these two days. According to records, a rain was detected on 14 May 2014 in the morning and in the evening. On 2 July 2014 a drizzle was detected in the morning and in the afternoon. It is important to stress that the rain sensor attached to the HATPRO instrument did not detect rain events during these two days. This fact leads to an important conclusion that it is not sufficient to control the observational conditions only at the radiometer site and that the data selection criteria used in the present study should be supplemented by additional requirements. For example, in the considered case the SEVIRI data showed unrealistically high values reaching 2 kg m$^{-2}$ and

the effective radius was also high (about 24 micrometres), which is either an error or the droplets were mainly raindrops. The quality flag of the SEVIRI results did not give a hint for errors but the cloud type was "supercooled".

Fig. 7 presents the examples of instantaneous measurements of LWP by SEVIRI and by HATPRO for several days of the CD season. One can see that in contrast to the WH season there are no cases with excellent agreement between the HATPRO and the SEVIRI. There is one case of good agreement 19 April 2013 which demonstrates the same qualitative and quantitative behaviour of LWP detected by the two instruments. The other cases in Fig. 7 display considerable differences reaching sometimes one order of magnitude. However despite large differences, the HATPRO and the SEVIRI observations provided the same qualitative behaviour of LWP on 17 April 2014 (one maximum of LWP in the morning) and on 21 April 2014 (one lower maximum of LWP in the morning and one higher maximum of LWP in the evening). The observation records at the meteorological station indicate the light snowfall on 8 April 2013 when SEVIRI produced extremely large values of LWP. It should be noted that the SEVIRI algorithm identified the clouds in this case as "supercooled". Rain was detected at the meteorological station on 23 February 2014 in the morning and in the afternoon, just before and after the period of LWP observations shown in Fig. 7. We stress, that the rain sensor attached to the HATPRO instrument did not detect rain events on this day.

It should be emphasized that, as one can see from Figs. 6 and 7, there is no preference on whether to perform averaging of the HATPRO data over 20 min period of time or over 60 min period of time. For the cases with good data agreement, both $HAT_{10-20}$ and $HAT_{10-60}$ datasets correspond well to the satellite data. For the cases with large discrepancies, the difference between the HATPRO and SEVIRI data is several times or even by the order of magnitude higher than the difference between the corresponding values of the $HAT_{10-20}$ and $HAT_{10-60}$ datasets.

Analysis of cases with very good data agreement gives the opportunity to estimate the influence of the LWP spatial gradients on the results of the comparison of the satellite and the ground-based data. As it has been mentioned in Section 3, the parallax effect for St.Petersburg measurement site is expected to be not less than 3.1 km in terms of the displacement to the North direction, i.e. the satellite image point corresponding to a cloud view by the HATPRO radiometer is located over the coastline or over the water of the Gulf of Finland. The long-term observations by SEVIRI revealed considerable difference of LWP over land and sea that means the strong inhomogeneity of the cloud distribution in the vicinity of the radiometer site. Under such conditions the parallax effect should be compensated in one way or another. This compensation can be done by the interpolation of the LWP values observed for pixels 0(243) and 221 (see Fig. 1). In order to obtain the rough estimation of the parallax effect and its compensation, we have plotted in Fig. 8 several examples of instantaneous measurements of LWP by SEVIRI for pixel 0, pixel 221 and the result of the linear interpolation of these LWP values to the parallax point. We compare these three quantities with the HAT10-20 dataset. First of all, it should be emphasized that in all selected cases the LWP values for pixel 221 were close to zero except the short period of time on 11 October 2014 and at the same time, the LWP values for pixel 0 were rather large and variable, that explicitly demonstrates the land-sea difference of LWP. The interpolated LWP values are lower than the values for pixel 0 except the short period of time on 11 October 2014. These interpolated values in general are closer to the corresponding values of the HAT10-20 dataset than the values observed for pixel 0. The improvement of the agreement after interpolation can be clearly seen on 6 May 2013 and on 6 June 2013 at the second maximum of LWP. On 5 October the interpolated values show excellent agreement with the HATPRO data most of the time excluding 3 hours in the afternoon when the SEVIRI data were oscillating. On 11 October one can also see certain improvement of the agreement between the HATPRO and SEVIRI data after interpolation. Table 4 presents the bias (HATPRO minus SEVIRI) and RMS difference between the LWP values derived from satellite and ground based observations for the cases shown in Fig. 8. In every case the interpolation resulted in considerable decrease of the bias absolute value. The RMS difference also decreased, however the effect is not so pronounced as for the bias.

# 5 Statistical LWP assessment

## 5.1 Seasonal features

We begin our analysis making a comparison of the instantaneous HATPRO and SEVIRI measurements of LWP by means of a two-dimensional histogram with the number of occurrence colour scale that is displayed in Fig. 9. This plot gives an impression about the overall agreement of measurements disregarding seasonal features. First of all, attention should be paid to the presence of a noticeable number of very high LWP values detected by the SEVIRI instrument and reaching 2.3 kg m-2. However, the number of occurrence of these measurements is very small if compared to the number of occurrence of the small values. The two-dimensional histogram for LWP<0.4 kg m-2 shown in the lower panel of Fig. 9 demonstrates that the largest number of occurrence is observed for small LWP not exceeding 0.03 kg m-2. The agreement between HATPRO and SEVIRI data for these values is good. For higher values, the agreement is not evident. This fact is not surprising since the agreement between instantaneous measurements is influenced by mistime, misdistance, weather conditions, type of cloudiness and the parameters of time averaging of the HATPRO data.

Now we analyse the conventional one-dimensional LWP frequency distributions in order to examine possible qualitative differences in HATPRO and SEVIRI measurements. The data were filtered to exclude the clear cases (LWP < 0.001 kg m-2) and the extreme cases (LWP > 0.4 kg m-2). It should be noted that data filtering has been done for collocated pairs of measurements: if one value in a pair was out of range, the whole pair was filtered out. The bin size has been selected as 0.02 kg m-2 and the frequency of occurrence was normalized with the total number of observations for the respective time interval. We show the distributions for the WH and the CD periods in Fig. 10 (1617 and 482 data samples correspondingly). Fig. 11 displays the monthly distributions for six months with the largest number of instantaneous measurements (200-500 values per month). February, August and October were not taken into consideration due to noticeably smaller number of data points.

The distributions for WH and CD periods of both, SEVIRI and HATPRO show, that the average LWP is low compared to LWP distributions that were averaged over the complete field of view of SEVIRI, also called "SEVIRI disc" (Kniffka et al., 2014). The distributions are lognormal, however for the CD period and the distribution has a bimodal structure, which is more pronounced for the results from SEVIRI. The secondary maximum for the distribution from SEVIRI is clearly identified at 0.12-0.14 kg m$^{-2}$ and reaches here about 13 % of the first peak. In case of HATPRO the secondary maximum is not well pronounced but it constitutes 20% of the first peak and is located at 0.10-0.12 kg m$^{-2}$. The maximum of distributions for WH and CD periods is in the bin 0.0-0.02 kg m$^{-2}$, for both HATPRO and SEVIRI. For the WH period, LWP frequencies quickly decline from the maximal number of occurrence of about 0.5 at low LWP values to smaller frequencies of 0.09 at LWP $\approx$ 0.05 kg m$^{-2}$. For the CD period the decline from the peak is more rapid for SEVIRI results: the frequency for LWP $\approx$ 0.03 kg m$^{-2}$ is already about 0.06, while for HATPRO results it is about 0.21.

Since the distributions obtained for seasonal periods differ considerably, we analyzed the monthly distributions also. In order to avoid misinterpreting of the results, we have chosen for our analysis only the months with the largest number of instantaneous measurements. As can be seen from Fig. 11, the bimodal structure of the distributions is detected for March, April, May and June (spring and early summer) with secondary maximum located at 0.10-0.14 kg m$^{-2}$. The distributions for July and September are mono-modal and resemble the seasonal distribution for WH period. The agreement between the SEVIRI and the HATPRO results is very good for all presented months of the WH period.

The distributions do not fall directly into one of the four categories in Kniffka et al. (2014), where all cloud types were characterised with mono-modal distributions, however they do resemble the low clouds category the most. The average all-

disc values range from 0.0672 to 0.0862 kg m$^{-2}$ (depending on the season) while the average HATPRO (SEVIRI) LWPs amount to 0.0182 (0.0274) and 0.0243 (0.0310) kg m$^{-2}$ for the cold, dry and warm, humid period. The climate of St.Petersburg is maritime where low stratiform clouds occur most frequently. Thicker, presumably convective clouds with LWP > 0.1 kg m$^{-2}$ form the secondary maximum in the distributions and occur in both periods (in the end of the CD period and in the beginning of the WH period).

## 5.2 Analysis of discrepancies

The distributions of SEVIRI's LWP is shifted to higher values in all months, particularly the secondary maxima are more pronounced than for HATPRO, the unfavourable observing conditions with a large viewing zenith angle of 72.48° cause large uncertainties. The root mean square error split into its systematic ($\sigma_s$) and unsystematic ($\sigma_u$) part following Anand et al. (1991) is displayed in Fig. 12 where only data points were taken into account where at least one of the both data sources provided LWP > 0. As can be clearly seen, the $\sigma_s$ dominates over the unsystematic fluctuations in all months; the average $\sigma_s$ is 0.07 kg m$^{-2}$ while the $\sigma_u$ amounts to 0.03 kg m$^{-2}$. The $\sigma_u$ stays relatively constant over the analysed time period with a standard variation (derived from the monthly $\sigma_u$-values) of 0.006 kg m$^{-2}$, however the $\sigma_s$ has a standard deviation of 0.053 kg m$^{-2}$. The month-to-month variation of $\sigma_s$ is about 9 times higher and exhibits a clear seasonal cycle with smallest values in February and March, then highest values from April to June and smaller values again from July to October. This result is unexpected because the summer months allow for the best viewing conditions for the SEVIRI and therefore the error should be smallest. However, the error could increase in summer due to high variability of convective clouds which reduces the representativeness of the HATPRO measurements for the SEVIRI pixel and complicates the comparison in contrast to situations with more stratiform conditions typically more frequent in winter. The detailed discussion of the problem of representativeness can be found in the paper by Slobodda et al. (2015). In this study, the SEVIRI retrieval produces some unrealistically high values of LWP mainly in the months April, May and June (up to 2.5 kg m$^{-2}$) which influence the RMSE to a large extent. In April, 1.1 % of the SEVIRI measurements showed LWP > 0.7 kg m$^{-2}$ which did not occur in the HATPRO measurements at all.

The sources of systematic and unsystematic discrepancies are multiple. They can be related to the retrieval algorithms, parameters of time-averaging of HATPRO data, viewing conditions of SEVIRI, and also to weather conditions, type, height, spatial and temporal evolution of clouds, and the magnitude of parallax effect. The analysis of the details of retrieval algorithms is beyond the scope of the present study. The variation of the averaging period was shown to have minor influence on the results of comparison. Therefore, while making the analysis we focused on weather and cloudiness conditions provided by the SEVIRI observations simultaneously with LWP data. The cases of the unrealistically high LWP values obtained by SEVIRI have been analyzed in detail and it has been found that the corresponding clouds are all of type "supercooled", the assigned cloud optical thickness value is quite often "100" and the effective radius of the droplets is rather big. The cloud top height did not show anything specific, clouds were between 2600 m and 9800 m. The quality mask revealed no abnormal situations: solar illumination was good, viewing conditions were fine, the input from numerical weather prediction showed no low level inversion and all measurement channels were present. On the basis of this information we suggest that the possible supercooled clouds with simultaneously very high effective radii can be the indication of the presence of erroneous retrieval results. According to the retrieval algorithm, clouds are marked as supercooled if the probability for being ice is lower than 0.5 and the temperature is below 273 K. One can suppose that our cases of unrealistically high LWP values obtained by SEVIRI are misclassified ice clouds. This idea is also in line with the high effective radii.

## 6 LWP diurnal cycle analysis

The diurnal cycle of LWP is an important characteristic which is necessary for numerical models since clouds have a strong influence on the earth's radiation budget. Both considered instruments are capable of registering the diurnal cycle. HATPRO operates day and night. The SEVIRI observations are limited by the condition $SZA < 72^o$, so for subarctic territories the observation period during a day differs greatly depending on season. Figs. 13 and 14 present the mean and the median LWP values as a function of a fraction of a day $F$ where the fraction of a day is a normalized period between $SZA = 90^o$ in the morning ($F = 0$) and $SZA = 90^o$ in the evening ($F = 1$). This period was divided into 10 sub-intervals. All LWP values less than $0.4 \, kg \, m^{-2}$ falling within each sub-interval during selected month of the years 2013 and 2014 were used as a source for calculations of mean and median values corresponding to a sub-interval. The reason for the given upper limit for LWP is the fact that the value $0.4 \, kg \, m^{-2}$ has been reported earlier as a threshold LWP between non-rainy and rainy atmosphere (Maetzler, 1992). Subintervals with the number of measurements less than 10 were excluded from analysis. We see that due to the limitations of the SEVIRI observations the shortest diurnal cycles are in February, March and October: 40 %, 60 % and 60 % of the daylight correspondingly. It is necessay to remind that the initial datasets consist of rain free days only, therefore the analysed diurnal cycles do not present the overall estimate but only the subset of purely liquid clouds during rain-free days.

First, we would like to pay attention to the fact that the comparison of median values gives an impression of the lack of agreement between the HATPRO and the SEVIRI data. At the same time, a good agreement is clearly seen for several months if the mean values are considered. For many cases the SEVIRI median LWP is lower than the corresponding HATPRO results and exactly equal to zero while HATPRO shows some variations of median LWP. We suggest two possible reasons for that: (1) the relatively low number of source data; (2) the underestimation of small LWP values by SEVIRI (zero LWP output in cases when HATPRO detects low LWP). So, we restrict further analysis only to mean LWP values and omit the word "mean" for simplicity.

For the cold and dry season, the LWP diurnal cycles obtained by the two instruments agree very well for March and April. For most sub-intervals, the discrepancy is less than the sum of standard deviations of the mean LWP values. For February, there is only a qualitative agreement – the decrease of LWP during the observational period was detected by both instruments.

For the warm and humid season, the LWP diurnal cycles obtained by the two instruments agree very well for four months: May, August, September and October. For June and July, the two instruments revealed very large differences in LWP in the first half of a day but showed similar LWP cycle for $F > 0.7$ and $F > 0.5$ for June and July correspondingly.

One can see that the detected LWP cycles differ from month to month. Some common feature can be noticed for the summer months: the SEVIRI instrument detected two maxima - the higher one at about noon and the lower one in the evening. Possible reason for the first maximum can be the developing convection. For other months it is difficult to propose any simple explanations of the LWP cycle and to conclude whether these cycles are typical for considered months or not. Concluding this section, we would like to emphasize the importance of taking into account the interannual variability of diurnal cycles. Our estimations of the interannual variability were based on the reanalysis data and are presented in Appendix 2. It has been shown that the average diurnal cycles calculated for the period of our study (2013-2014) noticeably differ from cycles obtained for the longer period 2003-2012. Since the temporal and spatial resolutions of the reanalysis data are considerably coarser than of the SEVIRI and HATPRO data, the direct comparison of diurnal cycles is not possible.

**7 Summary and conclusion**

Liquid water path is one of the key parameters of clouds urgently needed for a variety of studies relevant to climate modelling at Northern latitudes. The LWP measurements made by the ground-based microwave radiometer RPG-HATPRO (Radiometer Physics GmbH - Humidity And Temperature PROfiler) functioning at the measurement site of St.Petersburg State University, Russia, and made by the SEVIRI (Spinning Enhanced Visible and InfraRed Imager) satellite instrument over the area in the vicinity of St.Petersburg ($60^{o}$N, $30^{o}$E) have been compared. The geographical area under investigation can be considered as belonging to sub-Arctic region of Europe (the latitude range $50^{o}$-$70^{o}$). The time period of selected datasets spans two years (December 2012 – November 2014) excluding winter months, since the specific requirements to SEVIRI observations restrict measurements at Northern latitudes in winter when the solar zenith angle is too large.

The high quality of ground-based MW measurements has been taken as a main criterion for the data selection procedure. 210 rain-free observation days have been selected for the comparison. Purely liquid clouds have been considered, the control has been done using the cloud phase parameter of the SEVIRI data. The ground based and satellite data have been synchronised and divided into two datasets corresponding to two seasons: cold and dry (December-April) and warm and humid (May-October). The original data provided by the HATPRO instrument have been time-averaged in order to conform to the one-pixel measurement by the SEVIRI instrument.

The results of the comparison of the LWP values retrieved from the HATPRO and SEVIRI observations in the vicinity of St.Petersburg have shown the following:

1) There is no influence either of the sampling interval (10 s or 120 s) or of the averaging period (20 min or 60 min) of the original HATPRO data on the results of the HATPRO-SEVIRI data comparisons. (The given values of the averaging period correspond to the values of advection velocity of about 6 m s$^{-1}$ and 2 m s$^{-1}$.)

2) There are two site-specific features. First, the land-sea gradient of LWP is clearly revealed by the satellite observations. The magnitude of the land-sea difference for mean LWP values calculated for two-year period is about 0.040 kg m$^{-2}$ which is about 50 % relative to the mean value over land. The radiometer site is located close to the coast line of the Gulf of Finland in the area of large LWP spatial gradients. The parallax effect of the satellite observations has been estimated as about 3 km in terms of the displacement to the North direction. It can be compensated by the linear interpolation between two pixels of the SEVIRI measurements. Taking into account the estimated considerable values of the parallax effect and the land-sea LWP gradient, one can come to the conclusion that the combination of both in specific cases can influence the results of the comparison of the SEVIRI and the HATPRO data. The second site-specific feature is the high latitude location of the radiometer site and the large pixel size at this high latitude. That resulted in the lack of SEVIRI measurements in the cold and dry season.

3) Case studies of the instantaneous measurements revealed that possible reason of occasional very large discrepancies between the HATPRO and SEVIRI data can be local rain events in the vicinity of the radiometer site which are not detected by the rain sensor attached to the radiometer but which appear in the field of view of the satellite instrument. The SEVIRI algorithm misclassification of the ice clouds as supercooled water clouds can be another reason of discrepancies. Therefore, we focused on the analysis of the median instead of instantaneous values. The comparison of the daily median LWP values has demonstrated the RMS difference of 0.016 kg m$^{-2}$ for a warm season that is considerably lower than the RMS difference for a cold season which is 0.048 kg m$^{-2}$. The daily median values averaged over the datasets constitute 0.017 kg m$^{-2}$ and 0.02 kg m$^{-2}$ for WH and CD datasets correspondingly. The bias is very small and it is negative for the WH season and positive for the CD season.

4) The frequency distributions of both, SEVIRI and HATPRO show, that the average LWP is low compared to all-disc LWP distributions obtained by the SEVIRI instrument in previous studies. The distributions are lognormal and have a bimodal structure, which can be seen particularly in the months February and September. The distributions do not fall

directly into one of the four categories in Kniffka et al. (2014), where all cloud types where characterised with mono-model distributions, however they do resemble the low clouds category the most.

5) The systematic difference between LWP obtained by HATPRO and SEVIRI dominates over the unsystematic discrepancies in all months. The month-to-month variation of systematic difference exhibits a clear seasonal cycle with smallest values in February and March, then highest values from April to June and smaller values again from July to October. This result is unexpected because the summer months allow for the best viewing conditions for both, the HATPRO and SEVIRI and therefore the error should be smallest. In this study, the SEVIRI retrieval produces some unrealistically high values of LWP mainly in the months April, May and June (up to 2.5 kg m$^{-2}$) which influence the RMSE to a large extent. In April, 1.1 % of the SEVIRI measurements showed LWP > 0.7 kg m$^{-2}$ which did not occur in the HATPRO measurements at all. To our opinion, in order to further analyse the reasons of these systematic differences, it would be useful to combine the HATPRO and SEVIRI data with collocated LWP data produced by the AVHRR instrument. Though the LWP measurement over St.Petersburg site is made by AVHRR only once per day, the size of the ground pixel of AVHRR is smaller than of SEVIRI and this fact would be very helpful.

6) For the cold and dry season, the LWP diurnal cycles obtained by the two instruments agree very well for March and April. For February, there is only a qualitative agreement. For the warm and humid season, the LWP diurnal cycles obtained by the two instruments agree very well for four months: May, August, September and October. For June and July, the two instruments revealed very large differences in LWP in the first half of a day. Some common feature can be noticed for the summer months: the SEVIRI instrument revealed two maxima - the higher one at about noon and the lower one in the evening. Possible reason for the first maximum can be the developing convection. For other months it is difficult to propose any simple explanations of the LWP cycle and to conclude whether these cycles are typical for considered months or not.

7) In order to represent the diurnal evolution of LWP, the reanalysis data have been taken into consideration. The outputs of MACC and ERA-Interim datasets averaged over the period of 2003-2012 demonstrated similar daytime LWP evolution. However, averaging of the ERA-Interim data over the relatively short period of the present study (2013-2014), produced different result, pointing to the high interannual LWP changes which may mask an expected daytime evolution.

As a final conclusion, we briefly name the identified problems relevant to the comparison of HATPRO and SEVIRI measurements of LWP at high latitudes over the complex terrain which includes land and water areas. A more extensive database is needed for comparisons, especially for analysis of the cold and dry season in order to explain, in particular, the differences between the observational and reanalysis-based LWP diurnal cycles. Additionally, reasons for occasional very large discrepancies between HATPRO and SEVIRI data have still to be confirmed.

**Acknowledgements**

The operation of the RPG-HATPRO instrument was provided by the Research Centre GEOMODEL of St. Petersburg State University (http://geomodel.spbu.ru/).

**Funding**

Microwave radiometric measurements and data processing were supported by Russian Science Foundation through the project No. 14-17-00096. Ground-based and satellite data analysis was supported by Russian Foundation for Basic Research through the project No. 16-05-00681.

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

**Table 1:** The data selection steps and the designation of HATPRO datasets.

| Original data | Step 1. Quality control | Step 2. Selecting different sampling intervals | Step 3. Averaging over different time periods | Step 4. Synchronisation with SEVIRI data | HATPRO datasets for comparison with SEVIRI results |
|---|---|---|---|---|---|
| December 2012 – November 2014 (2 years) | Non-rainy days, SZA less than 72$^o$, uninterruptible data flow within every specific day. | 10 s | 20 min | The same time scale has been used for all datasets (15 min interval). | $HAT_{10-20}$ |
| | | | 60 min | | $HAT_{10-60}$ |
| | | 120 s | 20 min | | $HAT_{120-20}$ |
| | | | 60 min | | $HAT_{120-60}$ |

**Table 2:** Seasonal periods for comparison of HATPRO and SEVIRI data.

| Designation of a period | Time intervals | Number of days | Total number of days | |
|---|---|---|---|---|
| WH (Warm and Humid) | 1 May – 30 November 2013 | 47 | 120 | 210 |
| | 1 May – 30 November 2014 | 73 | | |
| CD (Cold and Dry) | 1 December 2012 – 30 April 2013 | 39 | 90 | |
| | 1 December 2013 – 30 April 2014 | 51 | | |

**Table 3:** The bias (SEVIRI minus HATPRO) and rms difference (kg m$^{-2}$) between the daily median LWP values derived from satellite and ground based observations. Correlation coefficient $r_c$ is also given.

| Season | Difference | HATPRO datasets | |
|---|---|---|---|
| | | HAT$_{10-20}$ | HAT$_{10-60}$ |
| WH | bias | -0.0004 | -0.003 |
| | rms | 0.016 | 0.014 |
| | $r_c$ | 0.91 | 0.88 |
| CD | bias | 0.002 | 0.002 |
| | rms | 0.048 | 0.049 |
| | $r_c$ | 0.66 | 0.64 |

**Table 4:** The bias (HATPRO minus SEVIRI) and RMS difference (shown in brackets) between the LWP values derived from satellite and ground based observations (kg m$^{-2}$) for the cases shown in Fig. 8.

| SEVIRI data | Date | | | |
|---|---|---|---|---|
| | 6 May 2013 | 6 June 2013 | 5 October 2014 | 11 October 2014 |
| Pixel 0 | -0.004 (0.013) | -0.009 (0.036) | -0.024 (0.077) | -0.019 (0.070) |
| Interpolation between pixels 0 and 221 | 0.003 (0.010) | 0.004 (0.029) | 0.002 (0.062) | -0.001 (0.069) |

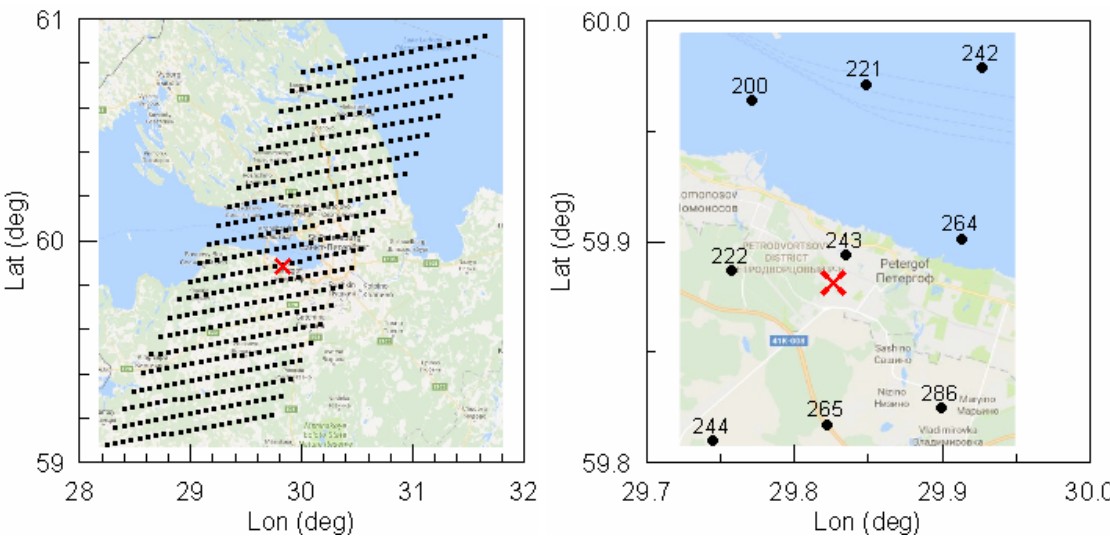

**Figure 1: The location of 441 SEVIRI measurement pixels (dots) selected for analysis (left, large terrain) and the location of the pixels nearest to the radiometric measurement site (right, small terrain, pixel numbers are shown). The position of the HATPRO radiometer is marked by the red cross. The distance from the centre of pixel 243 to the radiometer is equal to 1.5 km. The distance from the centre of pixel 242 to the radiometer is equal to 12 km.**

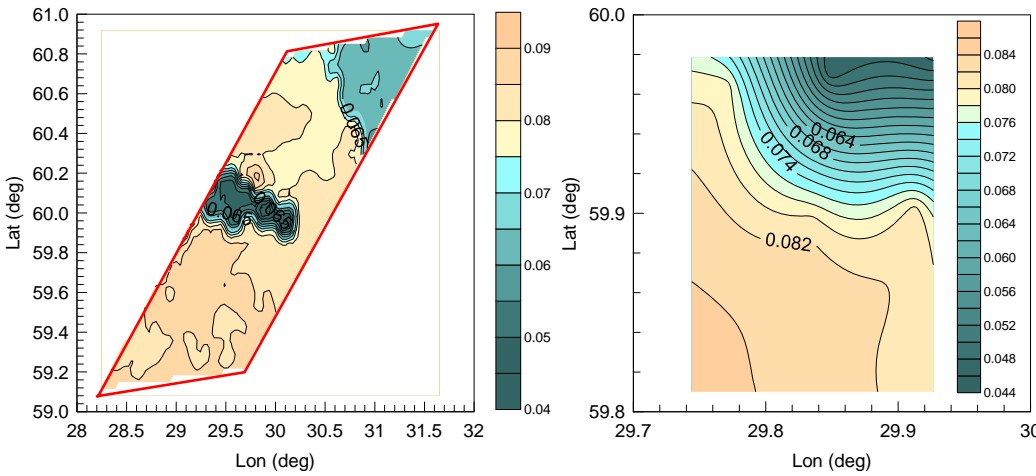

**Figure 2: The maps of the mean LWP values (kg m⁻², colour scale) calculated for the large terrain (left) and small terrain (right) and for the 2-year period 1 Dec 2012 – 30 Nov 2014 (measurements by the SEVIRI instrument).**

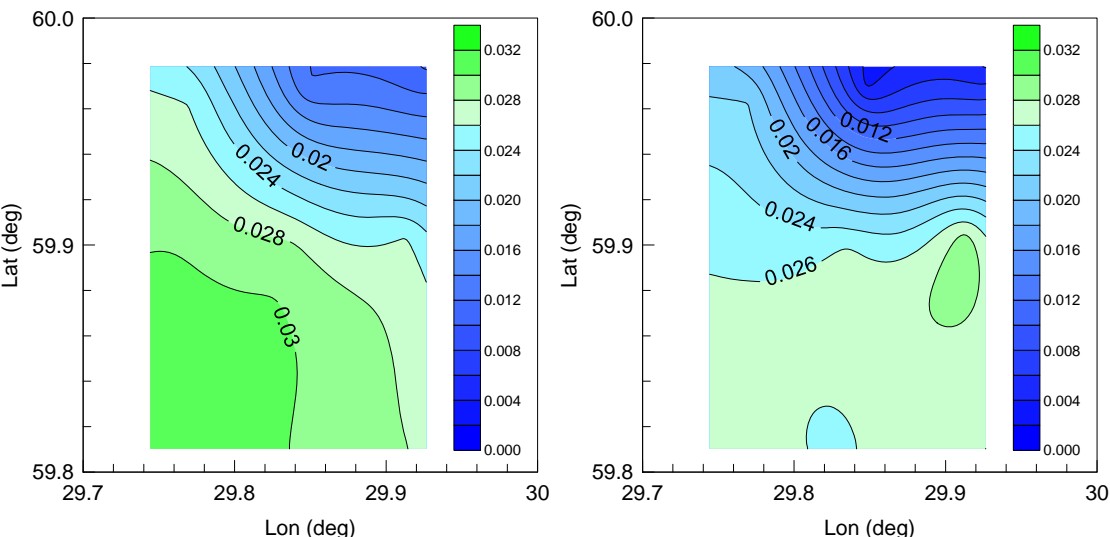

**Figure 3: The map of the mean LWP values (kg m⁻², colour scale) calculated for the small terrain and for the WH (left) and CD (right) datasets (measurements by the SEVIRI instrument).**

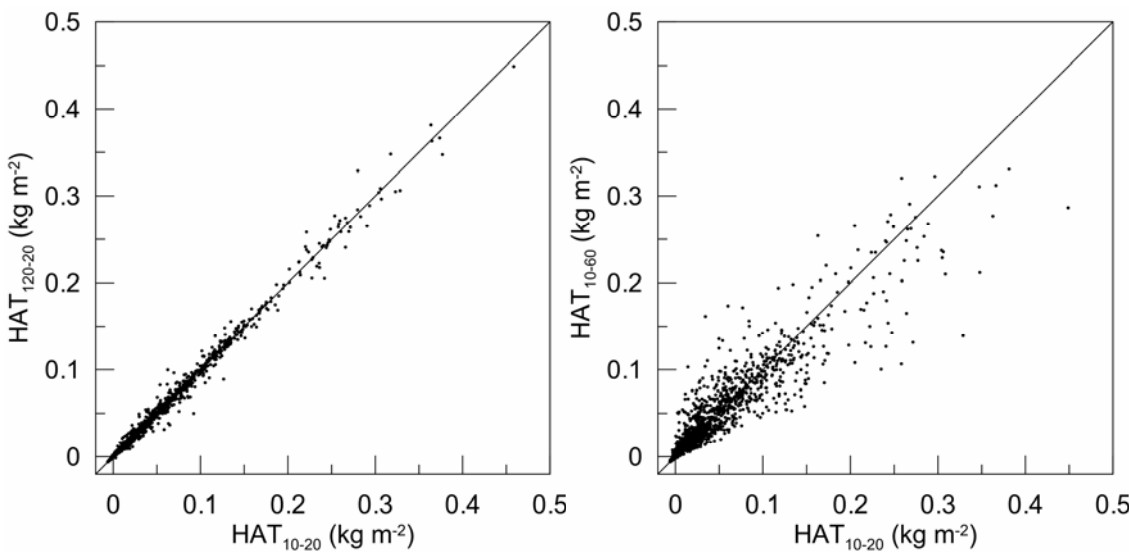

**Figure 4: The scatter plots of LWP data contained in different HATRO datasets, see Table 1. The left panel illustrates the influence of the sampling interval and the right panel illustrates the influence of the averaging period.**

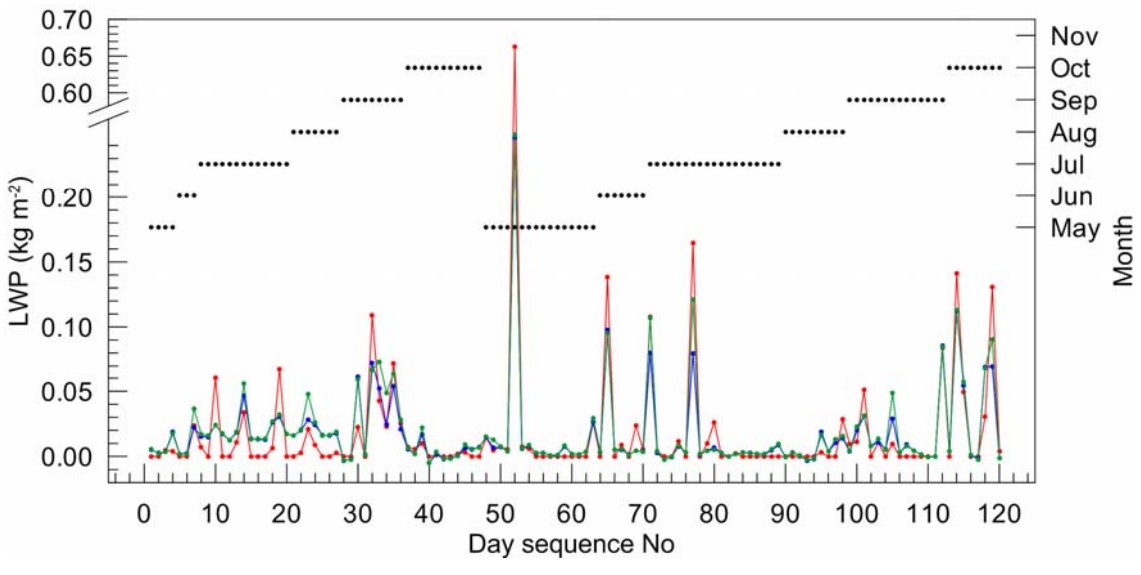

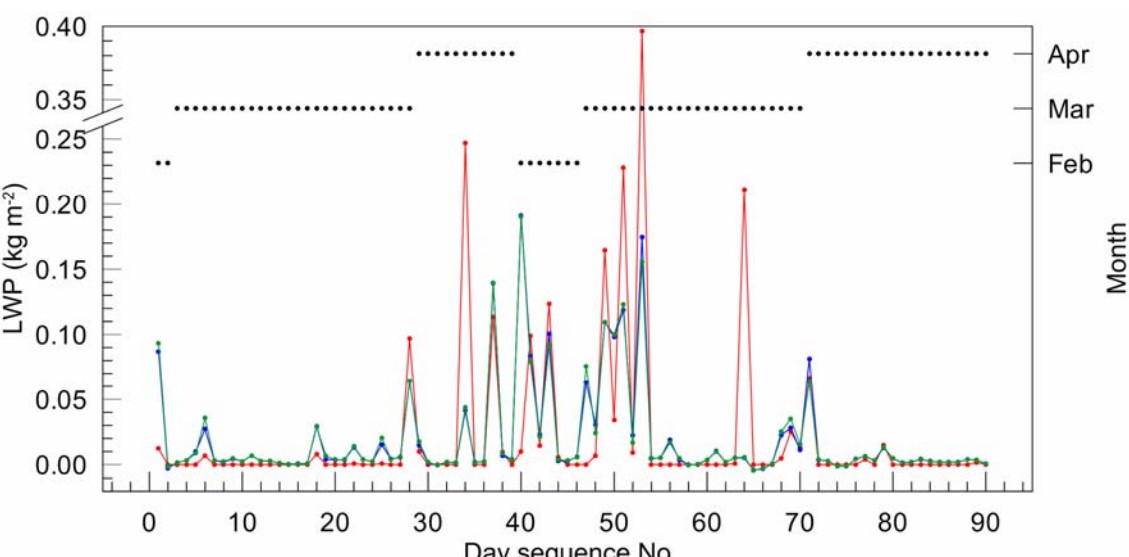

**Figure 5: The daily median LWP values obtained by SEVIRI (red dots) and HATPRO (blue dots for HAT₁₀₋₂₀ and green dots for HAT₁₀₋₆₀) as a function of day sequence number for the WH and CD seasons (top and bottom correspondingly). Colour dots are connected by lines only for demonstrative purpose. Black dots in combination with the right Y-axis indicate month of measurements.**

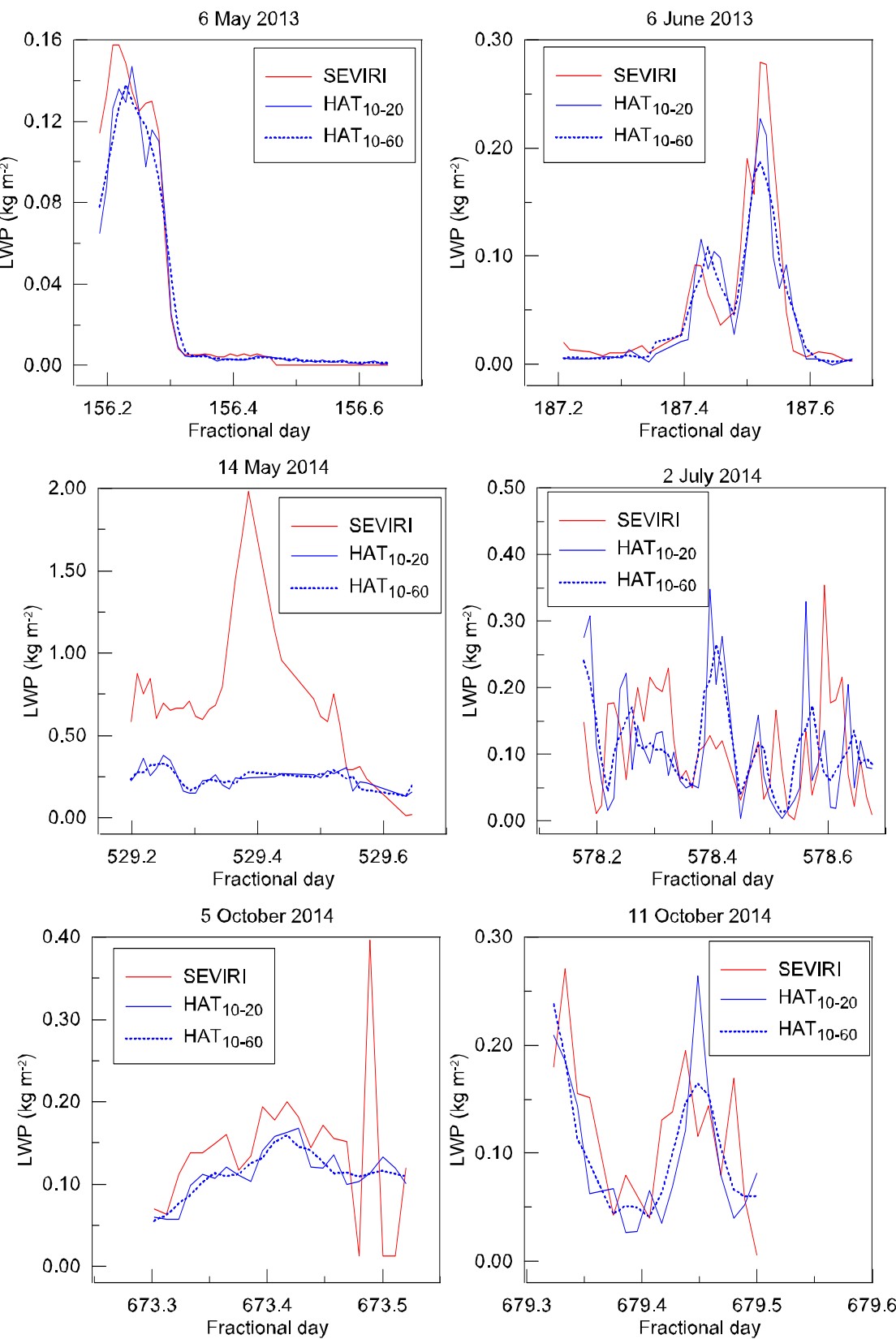

**Figure 6: The examples of instantaneous measurements of LWP by SEVIRI and by HATPRO (two HATPRO datasets used: HAT$_{10-20}$ and HAT$_{10-60}$). Several days of the WH season are shown (one day – one panel).**

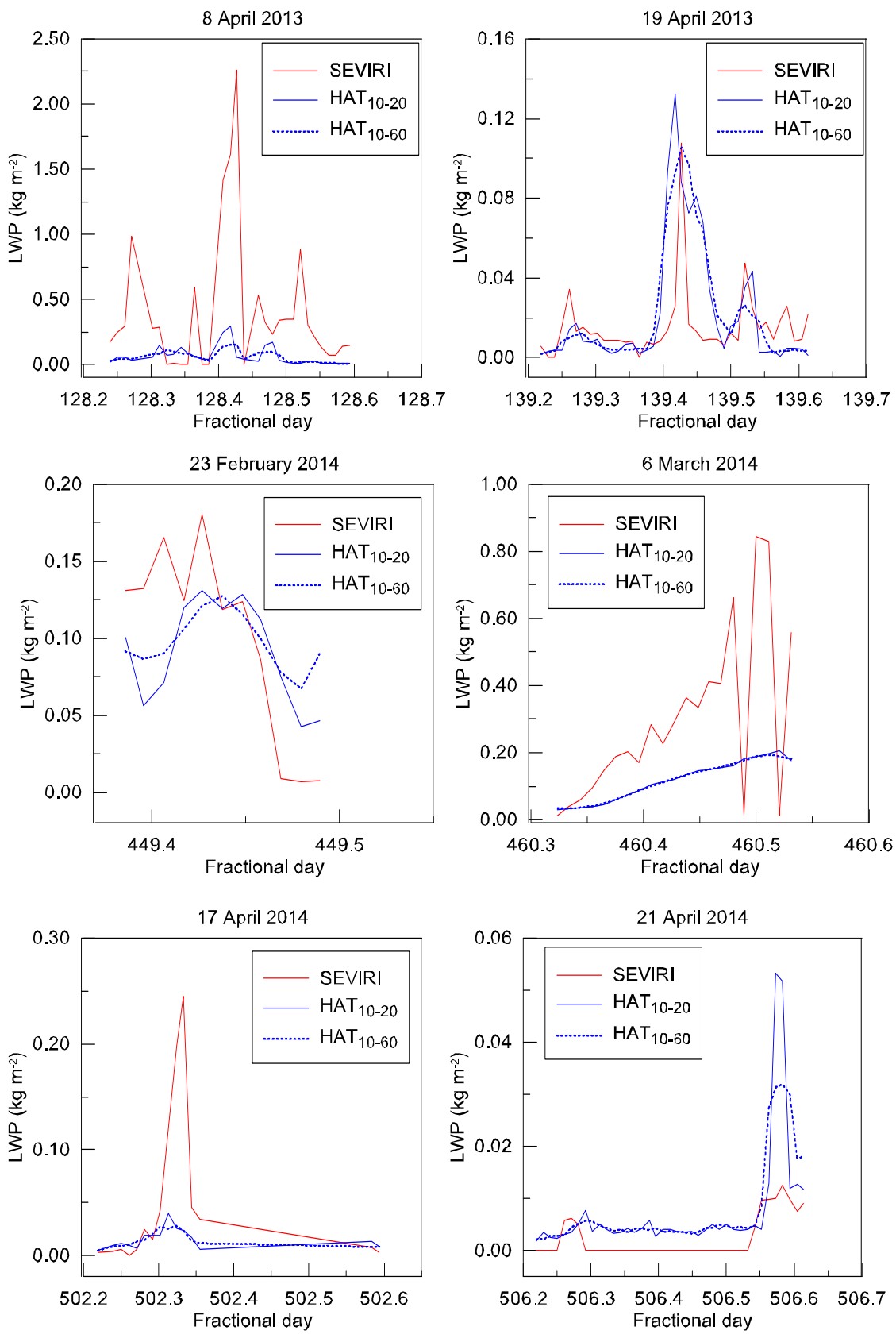

**Figure 7: The examples of instantaneous measurements of LWP by SEVIRI and by HATPRO (two HATPRO datasets used: HAT$_{10-20}$ and HAT$_{10-60}$). Several days of the CD season are shown (one day – one panel).**

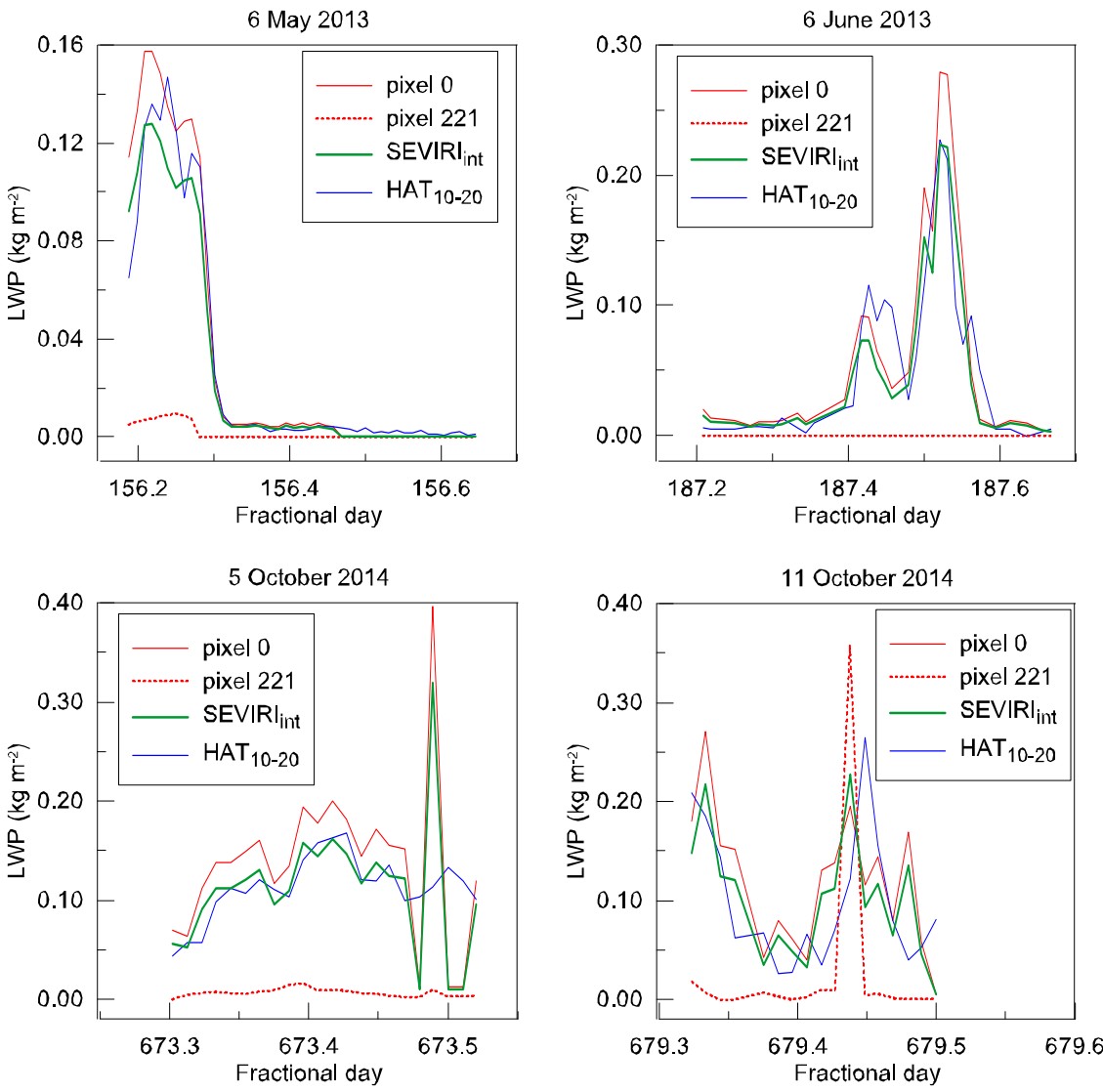

**Figure 8: The examples of instantaneous measurements of LWP by SEVIRI (pixel 0, pixel 221 and the interpolated value) and by HATPRO (HAT$_{10\text{-}20}$ dataset used). Several days of the WH season are shown (one day – one panel).**

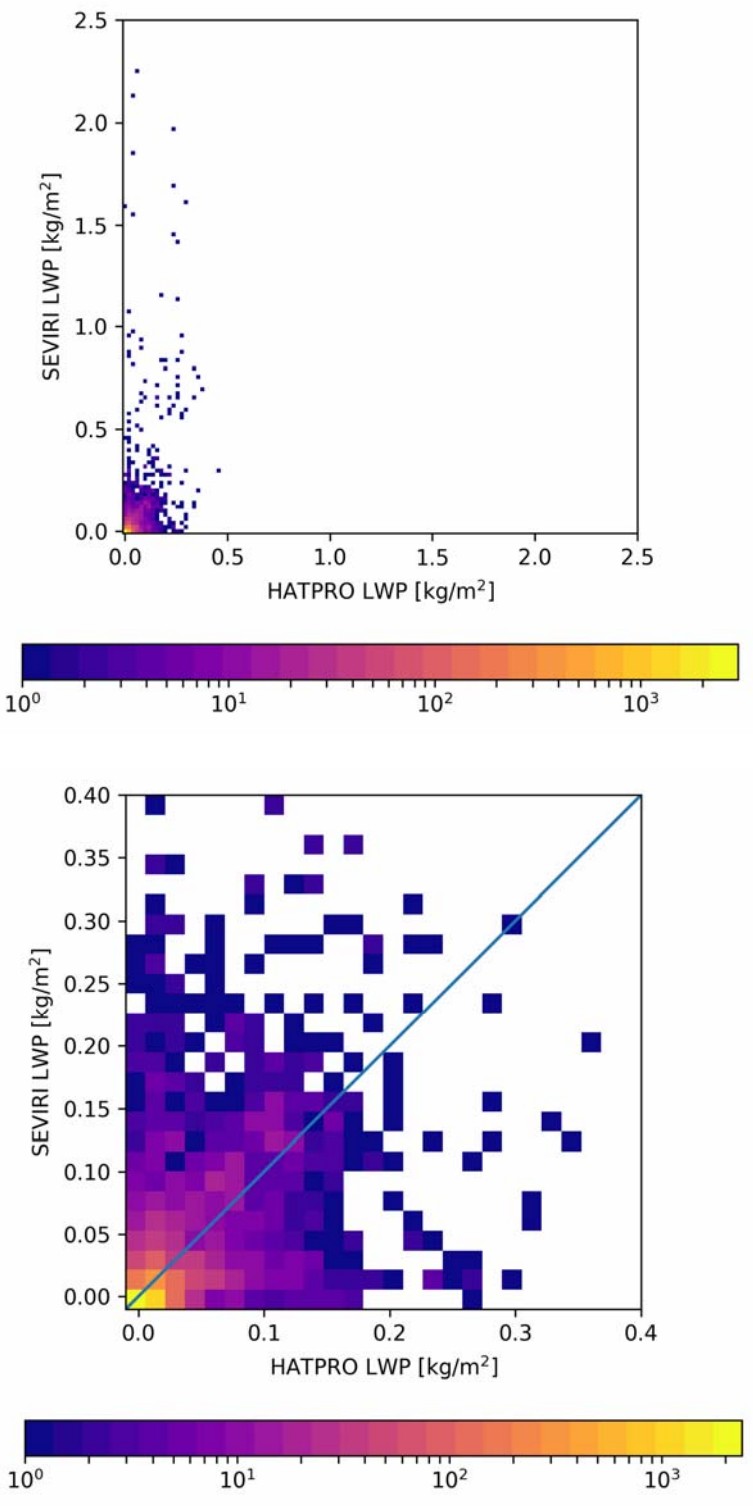

Figure 9: Comparison of the HATPRO and SEVIRI instantaneous measurements by means of two-dimensional histogram with number of occurrence colour scale. Upper panel: extra high LWP values are shown, lower panel: only LWP<0.4 kg m$^{-2}$ are shown.

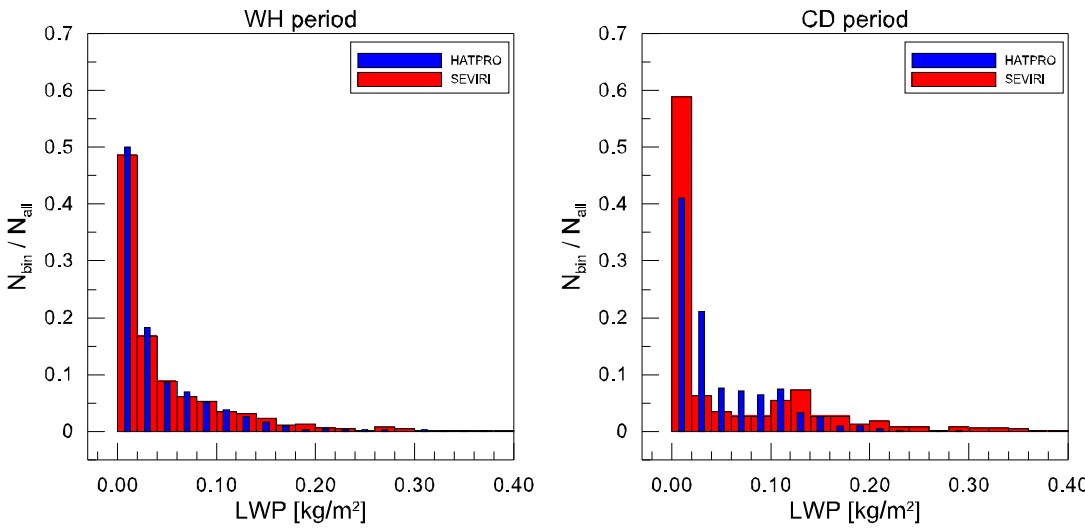

**Figure 10: Seasonal frequency distribution of LWP for HATPRO and SEVIRI normalised with the total number of occurrence: the WH period (left) and the CD period (right).**

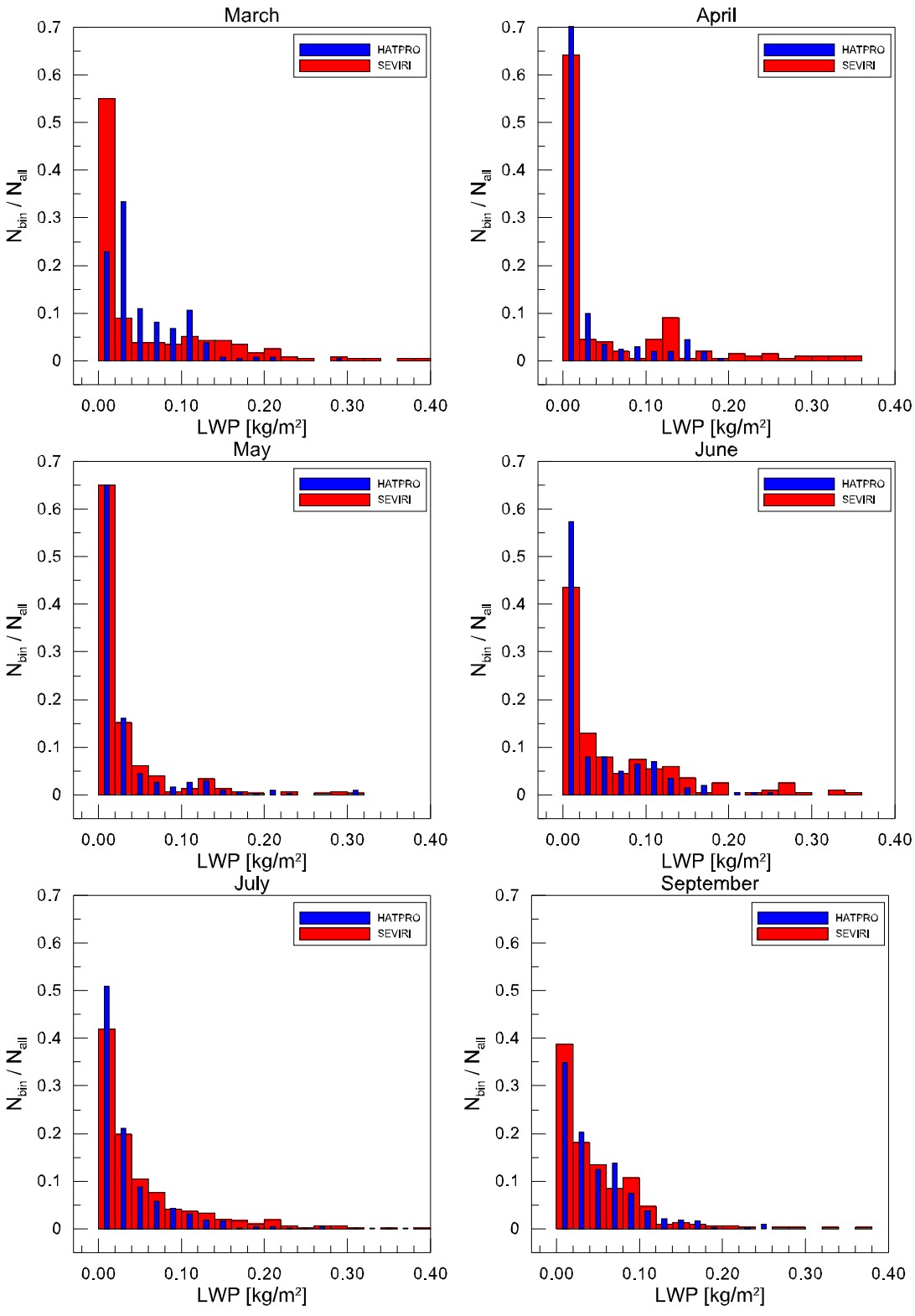

**Figure 11: Monthly frequency distribution of LWP for HATPRO and SEVIRI normalised with the total number of occurrence. Six months with the largest number of instantaneous measurements are shown.**

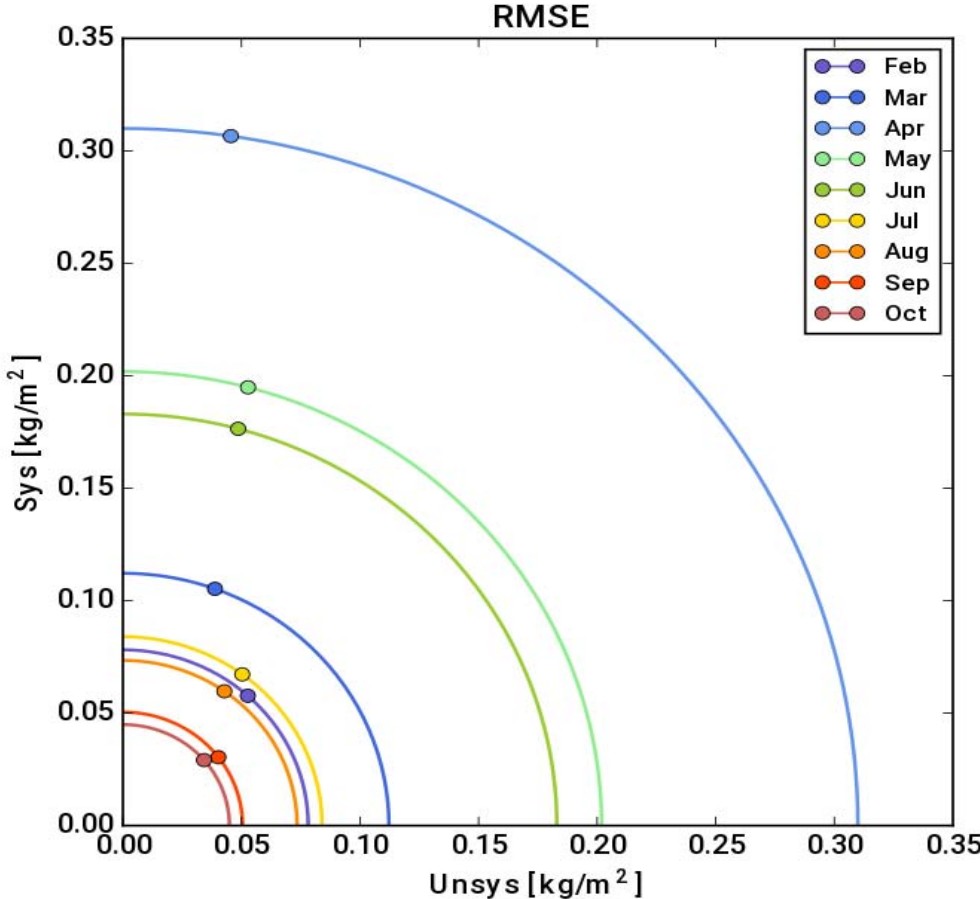

**Figure 12: Root mean square error divided into systematic and unsystematic parts for all months. The radii of the circles correspond to the monthly averaged RMSE values. Blue colours represent the cold and dry period, warm colours the warm and humid period.**

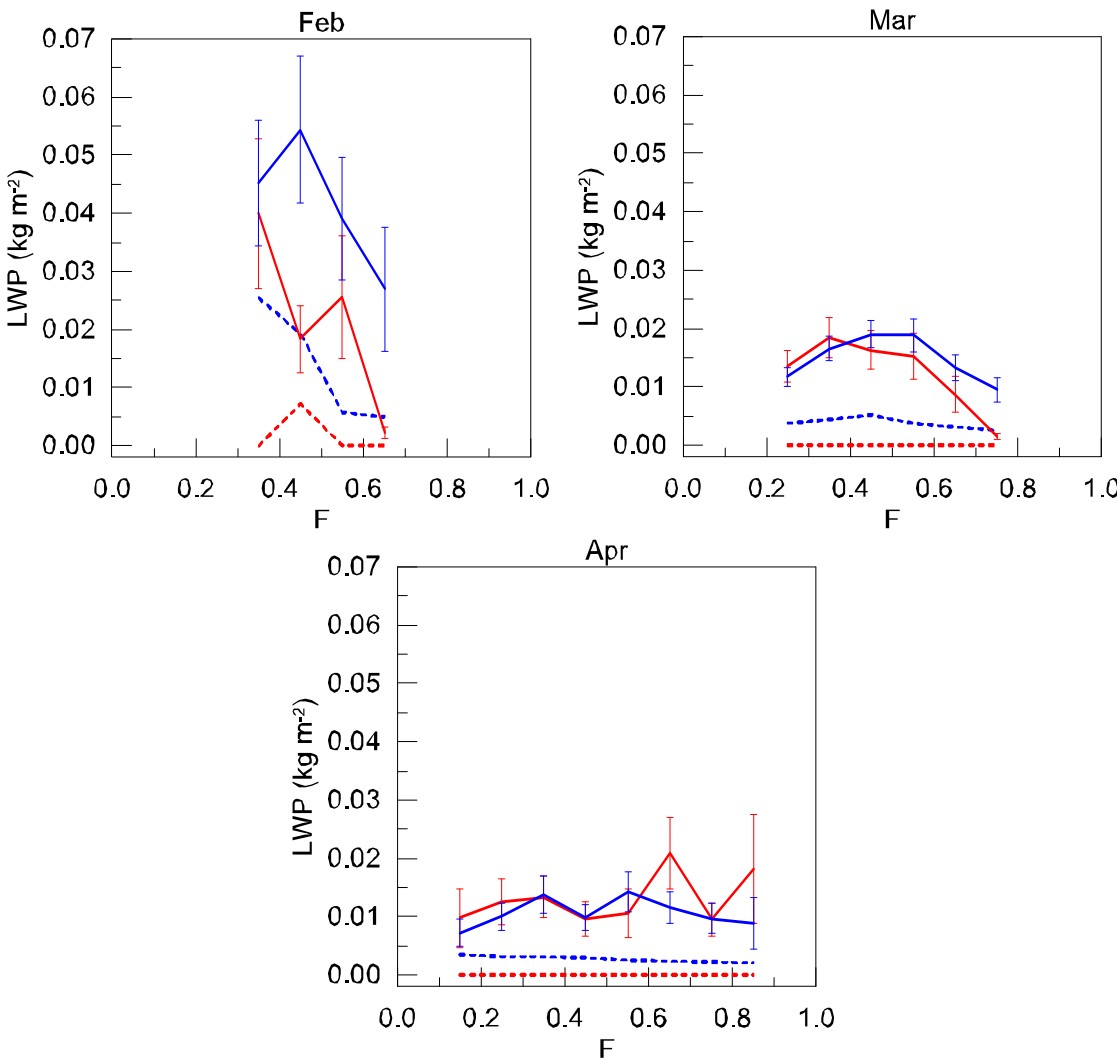

**Figure 13: The mean (solid) and the median (dashed) LWP retrieved from HATPRO (blue) and SEVIRI (red) as a function of a fraction of a day $F$ for February, March and April (the CD ensemble data), where the fraction of a day is a normalized period between SZA=90° in the morning ($F = 0$) and SZA=90° in the evening ($F = 1$). Error bars denote the standard deviation of the mean values.**

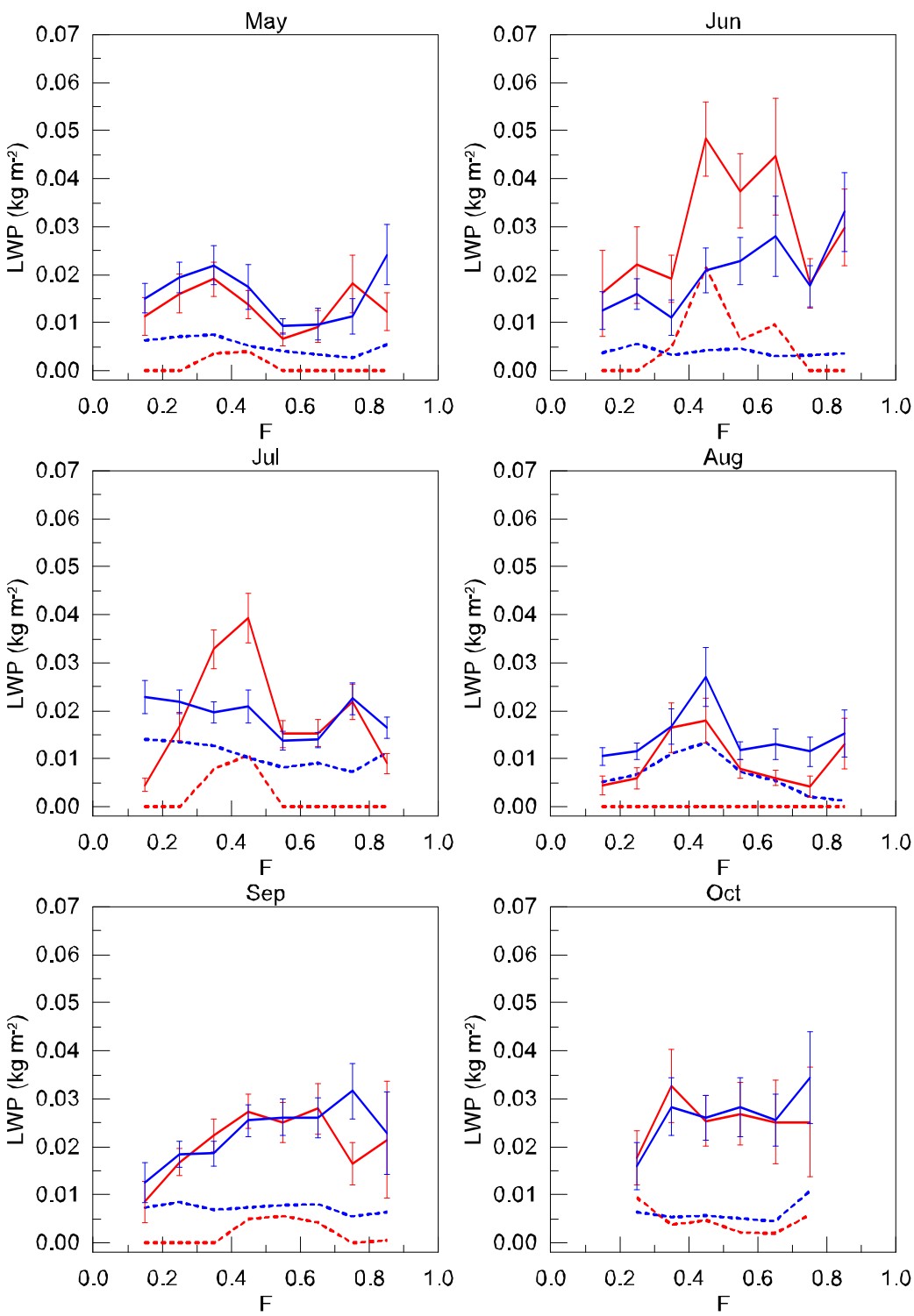

**Figure 14: The mean (solid) and the median (dashed) LWP retrieved from HATPRO (blue) and SEVIRI (red) as a function of a fraction of a day $F$ for May, June and July (the WH ensemble data), where the fraction of a day is a normalized period between SZA=90° in the morning ($F = 0$) and SZA=90° in the evening ($F = 1$). Error bars denote the standard deviation of the mean values.**

**Appendix 1 "The median LWP maps for the large and small terrains"**

Note: The median LWP value for CD period is equal to zero over the whole small terrain what means that clear sky conditions prevail in selected observations.

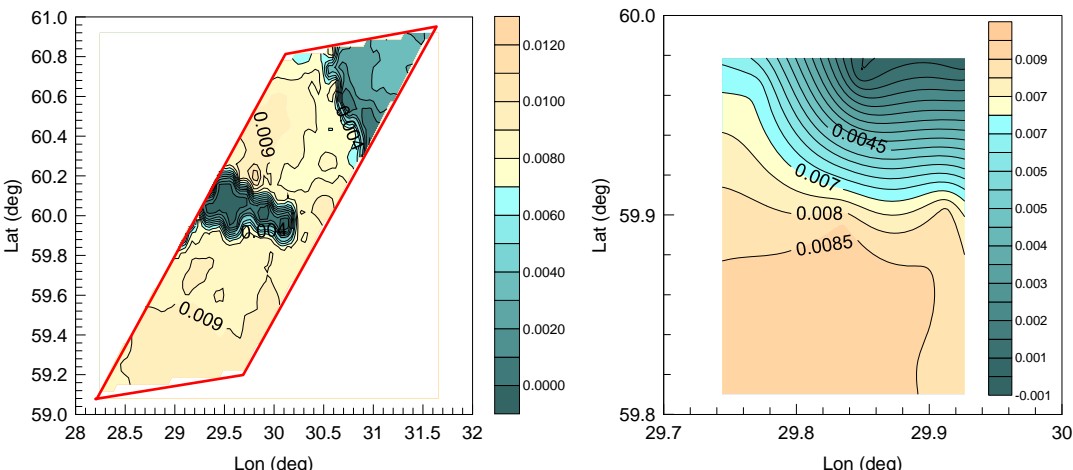

**Figure A1.1: The maps of the median LWP values (kg m$^{-2}$, colour scale) calculated for the large terrain (left) and small terrain (right) and for the 2-year period 1 Dec 2012 – 30 Nov 2014 (measurements by the SEVIRI instrument).**

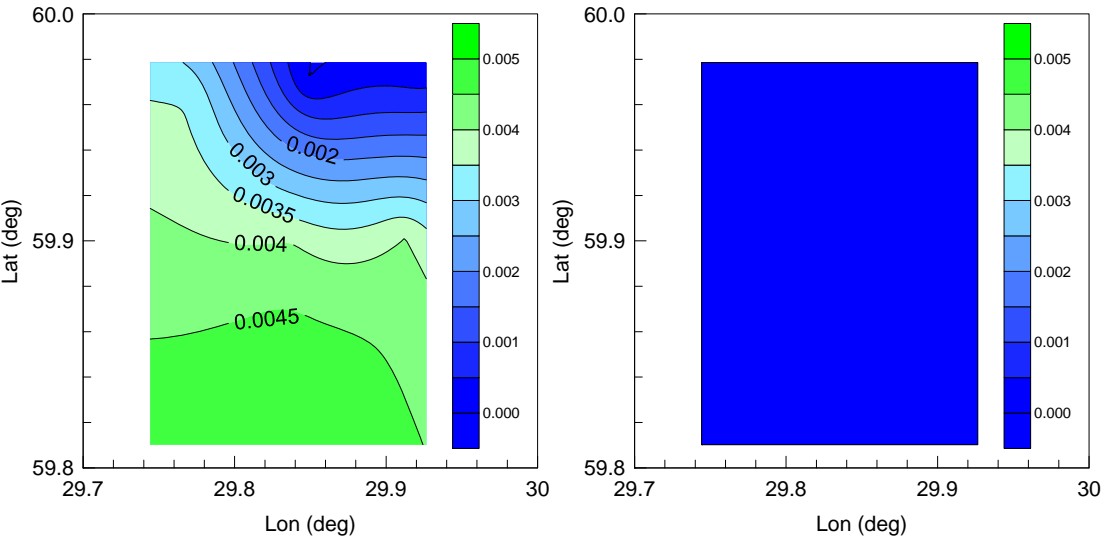

**Figure A1.2: The map of the median LWP values (kg m$^{-2}$, colour scale) calculated for the small terrain and for the WH (left) and CD (right) datasets (measurements by the SEVIRI instrument).**

**Appendix 2 "The diurnal cycle interannual variability as derived from reanalysis data"**

In order to evaluate possible diurnal cycles of LWP over St.Petersburg, multiannual data of reanalysis were considered. Two datasets were explored, ERA-Interim and MACC, both exploiting an assimilation of experimental observations and based on the European Centre for Medium-Range Weather Forecasts' (ECMWF, http://www.ecmwf.int/) Integrated Forecast System (IFS). ERA-Interim is the ECMWF reanalysis that covers the period from 1979 to the present time (Dee et al., 2011). MACC (Monitoring Atmospheric Composition and Climate) is a special reanalysis of atmospheric composition by assimilating satellite data into a global model, covering the period 2003-2012 (Innes et al., 2013). Both datasets are global, with a spatial resolution of ~80 km on 60 vertical levels from the surface up to 0.1 hPa. As MACC is limited to the time period of 2003-2012, the same year range was considered for the ERA-Interim dataset, for compliance. Reanalysis data were extracted over an area of 59.875-60.000°N / 29.750-29.875°E, enclosing the site of microwave radiometer observations near St.Petersburg. To represent the diurnal evolution of LWP, the outputs of MACC and ERA-Interim datasets were averaged over the period of 2003-2012 at a 3-hour time step (0, 3, 9, 12, 15, 18 and 21, universal time). Examples of derived average diurnal variations in April, July and October are shown in Fig. A2.1. In general, LWP in April is relatively low (~0.02 kg m$^{-2}$) with a weak variation during a day. Averaged over 2003-2012, LWP displays higher values (~0.06÷0.08 kg m$^{-2}$) and stronger variation in July and October, both in MACC and ERA-Interim datasets. Maximum LWP in summer (July) and in autumn (October) occurs in the early afternoon, roughly at ~15:00 local time (up to ~0.08÷0.11 kg m$^{-2}$). However, the absolute values of average LWP MACC and ERA-Interim data are somewhat different, with a distinctly higher amplitude of summer variation (July), derived from the data of MACC compared to ERA-Interim. Such amplitudes, calculated as a difference between the maximum and minimum LWP values for the months from February to October, are presented in Fig. A2.2. The amplitude of LWP diurnal variation averaged over 2003-2012 is higher in summer, with a maximum in June and July: ~0.04÷0.05 kg m$^{-2}$ and ~0.07÷0.09 kg m$^{-2}$ when derived from the data of ERA-Interim and MACC, accordingly. Thus, two reanalysis datasets (ERA-Interim and MACC) assume similar daytime LWP evolution over St.Petersburg, when averaged in 2003-2012: the maximum in the early afternoon and stronger in summer time (presumably due to the increase of convection). However, if one looks at the ERA-Interim data averaged over the period of our study (2013-2014), this conclusion is no longer so obvious (see Figure A2.1, bottom plot). Except in April, the average daytime LWP evolution in July and October, averaged over 2013-2014, is very different from the results of reanalysis of ERA-Interim and MACC over 2003-2012, and between each other: the July maximum is at 12:00, while the maximum in October is at 9:00. Besides, the amplitude of LWP diurnal variation is less in 2013-2014 compared to 2003-2012, with no maximum in summer (see Figure A2.2). To sum up, the exploration of reanalysis data over St.Petersburg reveals the presence of LWP diurnal variation, but at the same time points to the high interannual LWP changes which may mask an expected daytime evolution, when averaged over the relatively short period of 2013-2014.

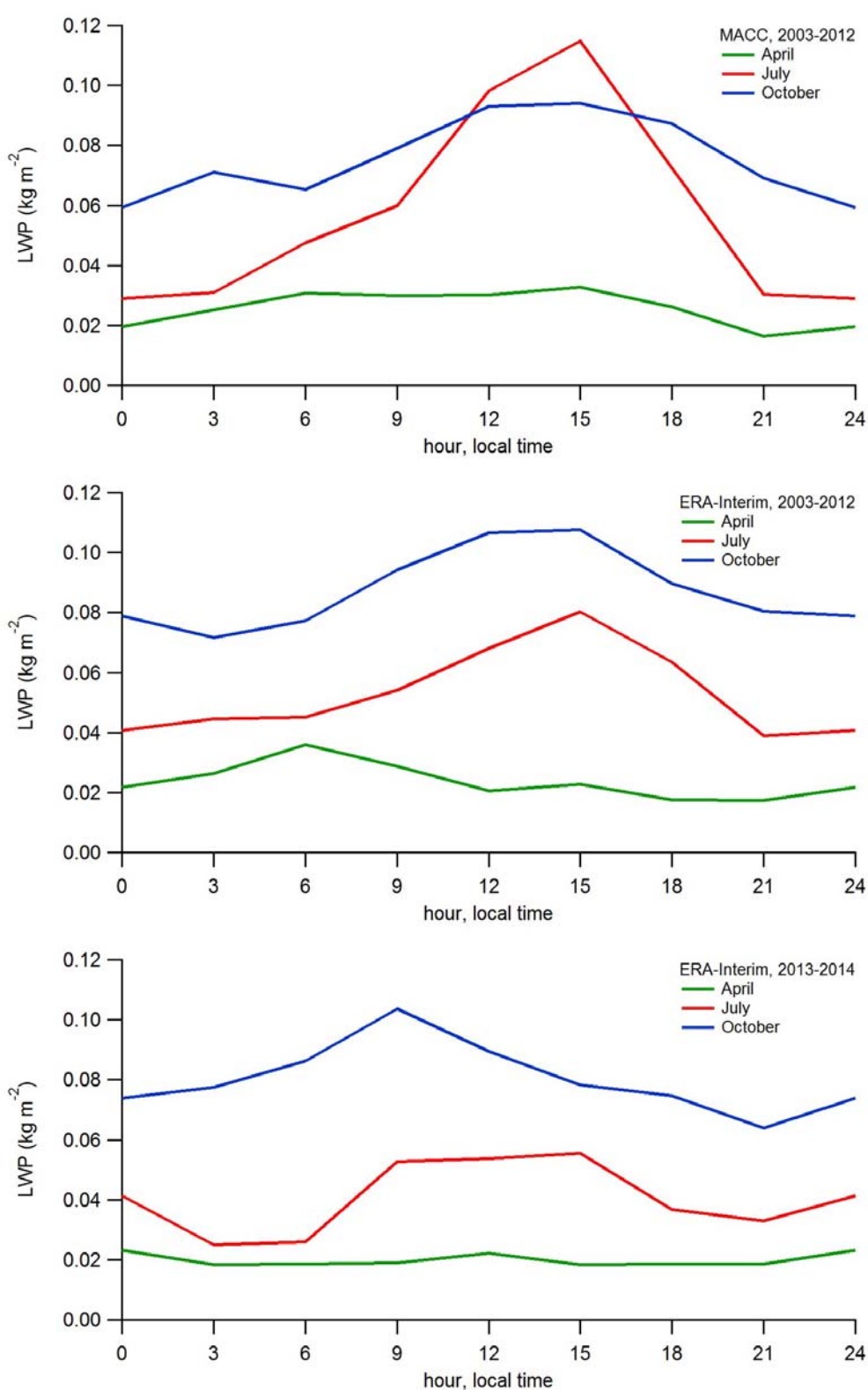

**Figure A2.1: The average diurnal LWP variations over St.Petersburg in April, July and October, derived from the data of reanalysis: MACC 2003-2012 (top), ERA-Interim 2003-2012 (middle) and ERA-Interim 2013-2014 (bottom).**

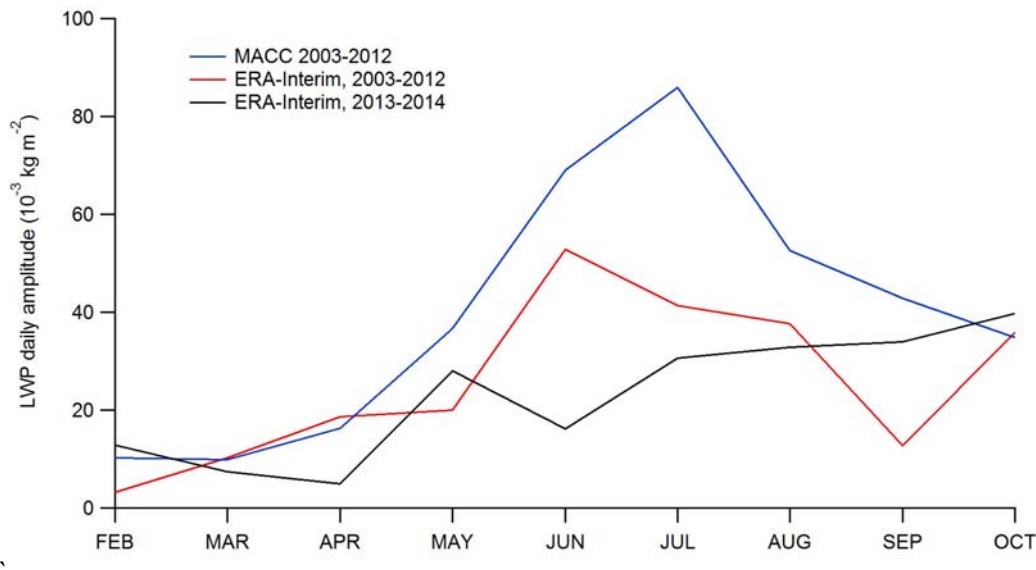

**Figure A2.2: The amplitudes of LWP diurnal variation, calculated as a difference between the maximum and minimum LWP values for each month of the year, derived from the data of reanalysis (MACC 2003-2012, ERA-Interim 2003-2012 and ERA-Interim 2013-2014).**