# Peer review of "Cloud liquid water path in the sub-Arctic region of Europe as derived from ground-based and space-borne remote observations"

_Atmospheric Measurement Techniques, 2018_

## Referee Comment (RC1) · D. Baumgardner (Referee) · 29 Jun 2018

This comparative evaluation of liquid water path (LWP) measured by satellite and ground based radiometers is a very well conceived and executed analysis that takes the reader step by step through the process of reconciling differences in an important property of clouds. The authors have carefully covered most of the important aspects of of such ana analysis that are needed to identify the differences and the potential sources that underlie the discrepancies.

I think that the intrinsic differences in measurement techniques, a detailed uncertainty analysis and the presentation of plausible reasons has been sufficiently provided, along

with a broad assortment of graphical representations that highlight the differences.

My only disappointment came when I was expecting to find the analysis put into the context of how important these differences are with respect to how they impact climate models since they begin the analysis by talking of the urgency of understanding how the lack of understanding clouds in this region is a major problem. When they began using the reanalysis data to look at diurnal cycles, I thought they would take the next logical step and either use a simple climate model to demonstrate the sensitivity of radiative forcing to differences in LWP, or at the least, test the statistical significance of the differences.

The lack of such a final analysis will not prevent me from recommending publication; however, providing some type of final analysis, either statistical testing or sensitivity analysis, I think would increase the scientific value of this paper.
* * *

---

## Referee Comment (RC2) · Anonymous Referee #3 · 13 Jul 2018

This paper is in the vein of similar studies comparing SEVERI derived LWP estimates with ground-based radiometer LWP measurements. However, this study concerns measurements in a very different and interesting region of Europe. The author's appear to have performed a through and careful analysis and the results should be of interest to the broader community.

I am in favour of publication with one broad caveat. Namely, the quality of the writing is quite uneven and must be addressed. The lack of indentations or spaces between paragraphs makes the manuscript harder to read than it should be. More seriously, the English usage in certain parts of the submission needs improvement. I do not have

**AMTD**

time edit the paper, here I will point out some examples ( I assume issues of this nature can be addressed during the revision process with the aid of a copy editor)

Line 7: "..cycle in Arctic.." ==> "cycle in the Arctic"

Line 25: "The interest to studies of the atmosphere in the sub-Arctic region is caused by the so-called "arctic amplification" effect that means the enhanced response of the arctic climate system to anthropogenic and natural impacts if compared to the response on the planetary average scale."

should be rewritten to something like:

"The interest of studying the sub-Arctic atmosphere is enhanced due to the so-called "arctic amplification" effect. This feedback effect is expected to enhance the response of the arctic climate system to both anthropogenic and natural forcing compared to the planet as a whole."

Line 31: "...particularly loss of.." ==> "..particularly the loss of ..."

etc..

---

## Referee Comment (RC3) · Anonymous Referee #4 · 14 Jul 2018

Liquid water path (LWP) is an essential climate variable which is rather difficult to measure. This results in a large spread of global mean LWP observations between 30m to 90 gm-2 which does not allow a proper evaluation of climate models today. Therefore, the investigation of LWP measured from the ground by microwave radiometry (MWR) and observed from geostationary orbit by SEVIRI at the site of St. Petersberg is of great interest. The study includes many interesting aspects but I would like to get a more solid statement to which degree one can rely on the different data sets.

While needing some editorial work, the manuscript is clearly written, addresses an important topic and presents interesting results which can have high impact on the

future observation network. Therefore, I recommend publication once the following points are addressed.

MAJOR POINTS

1) The manuscript needs to address the issue of LWP accuracy in more detail. This starts with emphasizing the lack of high quality measurements of LWP (being an essential climate variable), see for example the discussion by Lohmann and Neubauer (2018) who show that the global mean LWP varies between 30 and 90 gm-2 in the different global data sets. Most important, more information on the accuracy of the two LWP measurement techniques is needed. The manuscript frequently mentions the high quality of the ground-based microwave (MW) measurements but no quantitative values are given. Can they be used as a reference to estimate SEVIRI LWP accuracy? What are the advantages and disadvantages of both methods? What is their uncertainty? Do they have the same detection limit, i.e. I would expect SEVIRI to have higher sensitivity for low LWP values? I am wondering why the authors do not show the joint LWP distribution, i.e. two dimensional histogram with frequency of occurrence color code, which best illustrates the agreement of both data sets. The authors only provide the mean of WH (17 gm-2) and the RMS (16 gm-2) but do not make a statement that this would relate to an relative error of about 100%.

2) The LWP difference between land and sea for is strong and is one of the most interesting points of the paper. The paper takes it for granted that this is real but there needs to be a discussion/investigation whether this might be caused by a shortcoming of the SEVIRI, e.g. maybe due to the difference in surface albedo between land and sea. Furthermore, if it is true, a physical explanation for the LWP gradient needs to be provided. A potential explanation is the frequent presence of a high pressure system over the Baltic Sea and the associated subsidence which causes adiabatic warming and low cloudiness. With this explanation it might be better to separate the LWP time series into weather type situations rather than warm & humid (WH) and cold & dry (CD).

3) LWP statistics: LWP is highly variable in time and space and this variability strongly depends on the cloud type, i.e. is strongest for convective boundary layer clouds. Therefore, it is difficult to make solid statistics even if a two year data set is considered. By spitting the data further into individual months and climatic conditions the distributions become rather erratic and should not be overinterpreted. Smooth distributions require rather long time series (see Caddedu et al., 2013, Kniffka et al., 2014). Therefore, I recommend to just separate into the warm/humid and cold/dry regime or seasons at the most.

4) In respect to statistics the comparison with reanalysis is also difficult as only one instantaneous value every 3 hours is provided and thus only 8 per day and is not comparable with the better sampling of SEVIRI and MWR measurements. Thus it is the question whether the interannual variability shown add the end study is due to sampling or real and would require testing of the statistical significance. While I find it very important to make the point of high interannual variability I would remove the reanalysis aspect from the study as the data are not comparable in terms of spatial (80 vs 10 km) and temporal (8 to about 50) scales even with the coarser SEVIRI LWP data and also represent a mixed land & sea pixel.

5) The paper contains many plots and many could be eliminated. Why are s lengthy by showing both median and mean LWP. What is the benefit? The LWP distribution is strongly skewed towards low values and thus the median LWP values are typically lower than the accuracy. I would suggest to keep only the mean. If you would like to show the median then you could put it into an appendix.

M. P. Cadeddu, , J. C. Liljegren„ and D. D. Turner, 2013: The Atmospheric radiation measurement (ARM) program network of microwave radiometers: instrumentation, data, and retrievals. https://www.atmos-meas-tech.net/6/2359/2013/amt-6-2359-2013.pdf

MINOR COMMENTS:

L13: Provide also the relative error as the mean LWP is rather low and thus errors should often be in the order of 100 %.

L22: You just report on what you did but what was the result? Please extent.

L40: There are certainly many more studies on (sub)arctic clouds than the one by Garett and Zhao, the point to make here is that the measurement network is rather coarse in that region.

L52: Here, you need to emphasize on the importance of LWP as essential climate variable which is difficult to assess due to its high spatio-temporal variability, cf. Van Meijgaard and Crewell (2005) for the difficulties to compare LWP with models of different grid size.

L54: You need to explain the two satellite measurement principles for LWP from satellite and provide their limitations and uncertainties. 1) VIS/NIR observations only possible during daylight (not mentioned in manuscript) but available from geostationary satellite. This method needs to make assumptions on the vertical structure of the clouds as only the top can be sensed. 2) Microwave imagers on polar orbiters measure the emission signal which is only possible over the radiatively cold ocean - here you can cite Elsaesser et al. (2017) for the climatology and Greenwald et al. (2018) who investigates the uncertainty albeit by taking VIS/NIR as truth. Note, that due to the large footprints and the differences in emissivity between land and ocean no information for coastal pixels is available.

L55: I don't think it is necessary to list the satellite instrument names - if you do you need to provide the explanations for all acronyms.

L69: The uncertainty of ground-based MWR for LWP needs to be given

L87: Also the size of the footprint is larger than at the other locations

L92: "Since the LWP values can be essentially different over.." has this been shown in the literature or is this your result?

L95: Not give only one of the main goals but provide them all, ...for example the investigation of reanalysis quality...

L99: Why not cite Rose et al, 2005 at this stage - it is cited on 160

L129: 11km? The size at St. Petersberg needs to be mentioned

L144: It is not clear to me why only days which are completely free of rain are included in the comparison? This eliminates many data which are needed as the high variability of LWP makes statistical analysis difficult. If days with precipitation are excluded the mean diurnal cycle derived later on is only the mean of a subsample and not overall. At least this needs to be made clear maybe by giving it another name.

L146: How is a gap defined - a single 1 sec measurement?

L147: You speak about convergence - is the quality flag from the physical retrieval?

L160: In general, I think you need to explain why this is important: The retrieval algorithms are typically non-linear. Therefore, the retrieval needs to be made on high resolution brightness temperature data and subsequently then the LWP can be averaged but not vice versa. Note, that since the paper by Rose et al is already more than 10 years ago and since when resolutions of NWP have increased. It should be mentioned that situations can be very different and in particular convective boundary layer clouds have high variability.

L165: Please also write "stable" as this has nothing to do with the thermal stability but more with constant conditions. . L170: For a comprehensive discussion on how to compare LWP from ground-based observations and with spatial estimates from NWP models of different resolutions see Van Meijgaard and Crewell., 2005.

L180: Why boxcar? For simplicity?

L190: As Kostov et al. (2016) is not an open access paper the values for distinguishing WH and CD should be provided here. There is no surprise that WH is more frequent

as SEVIRI needs light.

L200: How good is the the geolocation of SEVIRI?

L203: especially as liquid clouds are at low heights.

L227: The statement is not correct - I still see the gradient.

L240: Different sampling and averaging times are chosen which provides some information on the error introduced by representativeness of the measurements. However, the optimal combination should depend on the actual weather situation - which should be discussed. Could the spatial distribution of SEVIRI or NWP might provide information to optimze this on an individual base.

L257: The agreement in daily mean LWP between SEVIRI and HATPRO in Fig. 7 is claimed "very good". I don't think so as many days especially in the beginning have zero LWP in SEVIRI and relevant values for HATPRO. Try to make a more quantitative statement here. You could also make a table for the daily means similar to table 4

L262: Is this for instantaneous or daily LWP?

L290: Section 2.2. needs to mention that SEVIRI retrievals fail in the case of strong vertical gradients and especially during rain. Why not use standard deviation of 3x3 SEVIRI pixels to identify (and eliminate) inhomogeneous situations?

L329 and Table 4: Why has this analysis not been done over the whole time period? It is important to do for generalization and it is a good transition to the next section.

L332: The title of the section should be changed to something like "Statistical LWP assessment". In this section I would have expected you - after looking at the mariginal distributions - to show the joint distribution which I am strongly missing here. This provides a much more direct view on the systematic and unsystematic components. In this sense L349: The name of section "5.1 Instrument differences" is a bit misleading. With this title I would have expected a discussion on the different sensitivities of the

instruments. I think no subsection headings are needed here. I recommend the authors to look at Tian et al. 2016 https://journals.ametsoc.org/doi/pdf/10.1175/MWR-D-15-0087.1 for a better error model.

L335: If you say "low compared to" but don't give any values this becomes difficult for the reader. It is also not possible to compare it with the full disk as many different climate zones are - better compare with other mid-latitude sites, cf. Cloudnet sites.. Her you should integrate the section 5.2 which somehow does not fit to the end. There a range for the distribution 0.2 - 0.6 kkgm-2 is given. This is better quantified by an interquartile or interpercentile range. Would be also good to give the numbers for the mean in both regimes. i

L340: I don't think that you can generalize the bimodal distribution for all February and September months with only two years and so much fluctuation in LWP. This is just due to certain weather patterns.

L340: Talking about 17% creates an impression of high accuracy - albeit the distributions are highly uncertain as they are determined by very view events and bimodality just comes by chance. You should provide the total number of samples for each distribution and as I suggest in the beginning just concentrate on the broader picture.

L345: Kniffka et al have a much larger and solid data base so that the comparison of the distributions is not fair here. Especially the bimodality (secondary maximum) is most likely just by chance and would not exist if a decade of data would be used - this could be checked by SEVIRI.

L355: Some background needs to be given on the method and an assessment of the "uncertainty".

L361: I would have expected the highest error in summer as the high variability of convective clouds reduces the representativeness of the HATPRO measurements for the SEVIRI footprint and complicates the comparison in contrast to situations with more

stratiform conditions typically more frequent in winter. cf. Slobodda, J., A. Hünerbein, R. Lindstrot, R. Preusker, K. Ebell, and J. Fischer, 2015: Multichannel analysis of correlation length of SEVIRI images around ground-based cloud observatories to determine their representativeness, Atmos. Meas. Tech. , 8, 567-578, doi:10.5194/amt-8-567-2015.

L360: As the authors also mention later on clouds with LWP of >0.4 kgm-2 are likely to precipitate and therefore I would suggest the authors to eliminate all SEVIRI retrievals above a certain threshold as the retrieval is not built for rain. Furthermore, it is only fair as rainy HATPRO measurements are eliminated as well.

L458: Why not mention the corresponding advection velocity?

L464: What is the accuracy of the 3.1 km estimate? I would simply say around 3 km.

L368: and the large footprint at this high altitude.

L479: Why do you know that it is log-normal? Did you test it? For this you probably need many more data so I would be more careful with the statement. Similar with the bimodality which is certainly due to poor sampling of different weather types and not a climate phenomenon.

Table 3: You also need to mention the number of samples, the mean value (so one can assess the relative error), bias corrected RMS and the correlation. Which SEVIRI pixel is taken?

Table 4: Not only the four cases but also the whole time period and WH and CD should be shown.

Figure 1: It is a bit irritating that pixel 242 is discussed but not shown.

Figure 3: The contouring provokes more interpretation than possible from only 3x3 points. Please plot them as blocks. Maybe you can even integrate them into the lowere left corner of fig. 2 as a zoom in. Same holds for all figures with 3x3 points, e.g. 4 and

[Figure]

5. Then you can also show WH and CD together.

Figure 6: You should give the number of samples, bias, rms, bias corrected RMS and correlation in a table similar to Table 3.

Figure13: Which averaging do you use here? I am again concerned with the statistics. Either leave this out as monthly data have poor statistics (give number) or if you want to do it keep it then use the full Taylor diagram. https://climatedataguide.ucar.edu/climate-data-tools-and-analysis/taylor-diagramspart

van Meijgaard, E., and S. Crewell, 2005: Comparison of model predicted liquid water path with ground-based measurements during CLIWA-NET. Atmos. Res., Special issue: CLIWA-NET: Observation and Modelling of Liquid Water Clouds, 75(3), 201 - 226, doi:10.1016/j.atmosres.2004.12.006.

Grammar, typos and reformulations

L10 - spell out SEVIRI and mention geostationary

L4: "have shown considerable differences" in you study (cf Fig.4) or in literature. Always make clear what are your results and what is known from the literature

L25: - this is not the only interest, I suggest: "There is increasing interest in the sub-Arctic region due to the so-called "arctic amplification" effect that .."

L28: "The large seasonal and interannual variation in low- and high-pressure systems and associated environmental variability due the location of the Baltic Sea between the North Atlantic and Eurasian air masses makes North Europe especially important to study.."

L57: "This study exploits the ... operating at..."

L65 Cloudnet not explained

L112:"in every detail" -> in detail

L125: Order of Dee and Dankers

L133: Channel "at" 1.6

L129: One sentence does not make a paragraph.

L171: Suggestion: Roebeling et al. (2008a) assumed Taylor's frozen turbulence hypothesis to relate an assumed wind speed of 10 m/s SEVIRI field of view (4 x 7 km2 resulting in a 20 min averaging period which Roebeling et al (2008b) extended to 30 min.

L182: The term synchonization does not fit so well - better use collocation or coincidences.

L184: CPH - all acronyms need to be explained period assuming the wind speed about 10 m s-1 and the).

L185: I think the argument goes into the wrong direction. The MWR gives you LWP in all situations but SEVIRI can not do so in case there is ice. Better write "Since SEVIRI cannot retrieve LWP..."

L201: Greuell and Roebeling (2009) studied the influence of the parallax effect (the horizontal displacement of a cloud viewed by SEVIRI due to its elevated height) on the comparison of SEVIRI and ground-based microwave LWP:

L285: rain

L354: "The coloured circles...." The figure caption does not need to be repeated.

L369: ..that the possible„

---

## Referee Comment (RC4) · Anonymous Referee #2 · 26 Jul 2018

The manuscript of Kostsov, Kniffka, and Ionov addresses the increasingly important task to merge spaceborne and ground-based observations for both, evaluation of retrieval techniques as well as for the characterization of the atmospheric state on medium and large scales. They are presenting a comparative study of liquid water path observations performed with microwave radiometer on ground and with SEVIRI aboard the Meteosat satellite. Given the high latitude of the investigated region of St. Petersburg they have to deal with significant parallax effects leading to ambiguities in the SEVIRI retrieval.

Overall, the manuscript is thoroughly structured, but some language editing is defi-

nitely required. I consider the scientific value as average, because the study does not go beyond a documentation of differences/similarities between ground-based and sat-based retrievals of LWP. It is only speculated about reasons for observed differences, and the authors do not point out any ways of omitting or correcting uncertainties in the retrievals. This however, does not prohibit to publish the manuscript because the information given seems to be well document.

Nevertheless, a few major and some minor comments remain which should be addressed before the editor (or a second round of review) will decided upon publication of the manuscript. Please see the comments below:

Section 2.1: This section should contain information about the calibration procedure of the HATPRO. How often where calibrations done? Is there any correlation between time since last calibration and quality of agreement between HATPRO and SEVIRI?

Line 159: Is sampling equal to averaging? If not, what is meant by 'sampling interval'? Does it mean taking just a single 1-s value every X seconds? If yes, why is this done? To safe disk space? To get into better agreement to the temporal resolution of SEVIRI?

Lines 195-200: Suddenly the 'Petergof' site is introduced, but where does it come from? Is this the location of the MW? If yes, it would fit to introduce Petergof in line 195: "...measurement site of Petergof."

Lines 262-263: Is there any possible explanation for the found extremely large discrepancy? Such a statement leaves a very curious reader. . ..

Line268ff: Which Seviri pixel was used for the comparison shown in Figure 8? Also at other positions in the text it was not always clear whether the shown results of SEVIRI are an average of the whole domain or only of a single pixel. If there was a general procedure applied, the authors should present it in Section 2.3.

Line 312: Is 'not less' the same as 'at least'? If so, at least would lead to less confusion while reading.
Line 483: What determines 'systematic' and 'unsystematic' discrepancies?

Lines 487-489, or Line 510: Can the authors present ideas, which parameters should be investigated to identify the reasons for the occasionally observed discrepancies? Would the analysis of rain radar data help? Or can it be attributed to strong small-scale fluctuations of LWP?

Line 165: situation .... situations

Line 166: form ... from

Line 215: superscript mˆ-2

Line 231: "…not one, but two SEVIRI pixels…."

Line 239: "…20 min and 60 min, respectively."

Line 346: mono-model ... mono-modal

Lines 406-407: "…for the summer months…"

…there are much more typos present in the manuscript…

Figures 2-5: The same scale should be used for all figures. At the moment, each figure shows a different range of values, making a visual comparison impossible. Also, it would be of interest, how many data points are contained in each map. Could the authors provide a map of the number of data points per pixel?

Figure 14: Please show a legend.
* * *

---

## Author Comment (AC1) · 28 Aug 2018

**The reply to D. Baumgardner (Referee, RC1)**

Dear Mr. Baumgardner,

We are grateful to you for the positive assessment of our study, for insightful remarks and for the valuable recommendations.

Below, your comments are given in **`bold courier font and blue colour`**.
The text added to the revised version of the manuscript is marked by red colour.

> **`My only disappointment came when I was expecting to find the analysis put into the context of how important these differences are with respect to how they impact climate models since they begin the analysis by talking of the urgency of understanding how the lack of understanding clouds in this region is a major problem.`**

We completely agree with this remark. When we started our investigation we also kept in mind the problem of the utilization of LWP data in climate models. However in the process of investigations we decided to focus only on the remote sensing aspect since quite a lot of problems relative to comparison of satellite and ground-based data have been identified.

> **`When they began using the reanalysis data to look at diurnal cycles, I thought they would take the next logical step and either use a simple climate model to demonstrate the sensitivity of radiative forcing to differences in LWP, or at the least, test the statistical significance of the differences.`**

The sensitivity study that you propose would be of course very interesting (we agree with you on that). But such a study does not fall into the scope of the present article which is devoted to the remote sensing aspects and to the problems of data comparison. We consider your recommendation as very valuable but we can not implement it as the part of the present study since it would require first the solution of the problems that have already been identified, and also the amount of data for such a study should be considerably increased.

> **`The lack of such a final analysis will not prevent me from recommending publication; however, providing some type of final analysis, either statistical testing or sensitivity analysis, I think would increase the scientific value of this paper.`**

As far as the statistical significance is concerned, we have the opinion that this task should be divided into two parts at least: the analysis of so-called "instantaneous" measurements and the analysis of the characteristics that are not much influenced by mistime, misdistance and averaging procedure: median values and frequency distributions. We have shown that for instantaneous measurements the analysis of the specific cases should be done, the statistics is not very helpful. However, in the revised version the new Figure is introduced which presents the two-dimensional histogram of the instantaneous measurements and corresponding analysis in Section 5.1:

[Figure]

Figure 9: Comparison of the HATPRO and SEVIRI "instantaneous" measurements by means of two-dimensional histogram with number of occurrence colour scale. Upper panel: extra high LWP values are shown, lower panel: only LWP<0.4 kg m$^{-2}$ are shown.

"We begin our analysis making a comparison of the instantaneous HATPRO and SEVIRI measurements of LWP by means of a two-dimensional histogram with the number of occurrence colour scale that is displayed in Fig. 9. This plot gives an impression about the overall agreement of measurements disregarding seasonal features. First of all, attention should be paid to the presence of a noticeable number of very high LWP values

detected by the SEVIRI instrument and reaching 2.3 kg m$^{-2}$. However, the number of occurrence of these measurements is very small if compared to the number of occurrence of the small values. The two-dimensional histogram for LWP<0.4 kg m-2 shown in the lower panel of Fig. 9 demonstrates that the largest number of occurrence is observed for small LWP not exceeding 0.03 kg m$^{-2}$. The agreement between HATPRO and SEVIRI data for these values is good. For higher values, the agreement is not evident. This fact is not surprising since the agreement between instantaneous measurements is influenced by mistime, misdistance, weather conditions, type of cloudiness and the parameters of time averaging of the HATPRO data."

---

## Author Comment (AC2) · 28 Aug 2018

**The reply to the anonymous referee #3 (RC2)**

We are thankful to the referee for the comments. We appreciate all the comments; we took them into account while preparing the revised version of the manuscript.

Below, the actual comments of the referee are given in **bold courier font and blue colour**. The text added to the revised version of the manuscript is marked by red colour.

> **This paper is in the vein of similar studies comparing SEVERI derived LWP estimates with ground-based radiometer LWP measurements. However, this study concerns measurements in a very different and interesting region of Europe. The author's appear to have performed a through and careful analysis and the results should be of interest to the broader community.**

We are grateful to the referee for the positive assessment of our study.

> **I am in favour of publication with one broad caveat. Namely, the quality of the writing is quite uneven and must be addressed. The lack of indentations or spaces between paragraphs makes the manuscript harder to read than it should be. More seriously, the English usage in certain parts of the submission needs improvement. I do not have time edit the paper, here I will point out some examples ( I assume issues of this nature can be addressed during the revision process with the aid of a copy editor)**

We completely agree with the referee that the lack of indentations and spaces between paragraphs can be annoying. However we prepared the manuscript in strict accordance with the template provided by the Journal. In the revised version we slightly violated the rules of the template and the spaces between paragraphs are present now. As far as the general comment about the language is concerned, we can say that we checked the text and did our best to improve the language of the revised version of the paper.

> **Line 7: "..cycle in Arctic.." ==> "cycle in the Arctic"**

Corrected.

> **Line 25: "The interest to studies of the atmosphere in the sub-Arctic region is caused by the so-called "arctic amplification" effect that means the enhanced response of the arctic climate system to anthropogenic and natural impacts if compared to the response on the planetary average scale."**
>
> **should be rewritten to something like:**
>
> **"The interest of studying the sub-Arctic atmosphere is enhanced due to the so-called "arctic amplification" effect. This feedback effect is expected to enhance the response of the arctic climate system to both anthropogenic and natural forcing compared to the planet as a whole."**

Corrected.

> **Line 31: "...particularly loss of.." ==> "..particularly the loss of ..." etc..**

Corrected.

---

## Author Comment (AC3) · 28 Aug 2018

**The reply to the anonymous referee #4 (RC3)**

We would like to express our gratitude to the referee for the very careful reading of our manuscript and for the helpful criticism. We appreciate all the comments; we took them into account while preparing the revised version of the manuscript.

Below, the actual comments of the referee are given in **`bold courier font and blue colour`**. The text added to the revised version of the manuscript is marked by red colour.

**`MAJOR POINTS`**

**`1) The manuscript needs to address the issue of LWP accuracy in more detail. This starts with emphasizing the lack of high quality measurements of LWP (being an essential climate variable), see for example the discussion by Lohmann and Neubauer (2018) who show that the global mean LWP varies between 30 and 90 gm-2 in the different global data sets. Most important, more information on the accuracy of the two LWP measurement techniques is needed. The manuscript frequently mentions the high quality of the ground-based microwave (MW) measurements but no quantitative values are given. Can they be used as a reference to estimate SEVIRI LWP accuracy? What are the advantages and disadvantages of both methods? What is their uncertainty? Do they have the same detection limit, i.e. I would expect SEVIRI to have higher sensitivity for low LWP values? I am wondering why the authors do not show the joint LWP distribution, i.e. two dimensional histogram with frequency of occurrence color code, which best illustrates the agreement of both data sets. The authors only provide the mean of WH (17 gm-2) and the RMS (16 gm-2) but do not make a statement that this would relate to an relative error of about 100%.`**

We addressed the issue of LWP retrieval accuracy in more detail. First, the reference to the paper by Lohmann and Neubauer (2018) has been included in the Introduction section and the importance of LWP accuracy assessment has been noted:

"It should be emphasized that LWP is an essential climate variable and the assessment and improvement of the accuracy of LWP data obtained from different platforms and instruments is still an actual problem. Lohmann and Neubauer (2018) have reported that global annual mean LWP values over oceans derived from measurements by different satellite sensors have very broad range of $30 - 90$ g m$^{-2}$; besides, both retrievals from visible–near-infrared sensors and microwave sensors have biases in LWP data. The validation campaigns for LWP measurements from space often use ground-based LWP observations by microwave radiometers as the reference data since they have a precision that is superior to current satellite remote sensing techniques (Roebeling et al., 2008a)."

Second, the quantitative values of the LWP retrieval accuracy by ground-based MW radiometers are given in the revised version in Section 2.1:

"The estimations of the accuracy of LWP retrievals by the HATPRO radiometer near St.Petersburg have been made previously (Kostsov et al., 2017) on the basis of the analysis of cloud-free situations and on the basis of calculations of the error matrix of the physical algorithm. The analysis of cloud-free situations has shown 0.009-0.011 kg m$^{-2}$ for bias and 0.001 kg m$^{-2}$ for random error. It should be noted that the corresponding values reported in the study by Matzler and Morland (2009) are 0.002-0.005 kg m$^{-2}$ and 0.001 kg m$^{-2}$. The

error matrix calculations have shown that the random error varies in the range 0.001 – 0.008 kg m$^{-2}$ for all observed LWP values (up to 1 kg m$^{-2}$). Cossu et al. (2015) obtained the slightly higher bias of LWP retrievals by ground-based MW radiometry which constituted 0.01-0.02 kg m$^{-2}$ and they also estimated the random error as 10-20% for LWP greater than 0.1 kg m$^{-2}$."

The information on the accuracy of LWP retrievals from SEVIRI observations is given in the revised version in Section 2.2:

"In the validation document of CM SAF (Finkensieper et al. 2016), the bias of the LWP measurements is specified to be 0.00007 kg/m² for monthly mean values compared to MODIS and the bias-corrected root mean square error amounts to 0.0101 kg/m². Here the complete field of view of SEVIRI and the monthly means from 2004 – 2015 were analysed. A comparison with AMSR-E was also conducted and showed a bias of 0.0034 kg/m² and a bias-corrected root mean square error of 0.034 kg/m². Unfortunately, this comparison was based only on a single overpass of AMSR-E. In Roebeling et al. (2008a) comparisons were made for the three sites Cabauw (Netherlands), Chilbolton (United Kingdom) and Palaiseau (France) for time-series of 4 years. Here the bias was found to be 0.005 kg/m² in summer and 0.010 kg/m² in winter while the variance was stable with 0.030 kg/m². Please note that the latter study was based on a retrieval algorithm state 10 years ago, until today, the retrieval has undergone many modifications that led to an overall improvement."

As far as the question of advantages and disadvantages of the methods is concerned, we have the impression that the answer is evident. However, in order to make the article clear to potential readers who are far from remote sensing problems we included small paragraphs in Section 1:

"Along with the high accuracy of LWP retrievals, other advantages of the ground-based MW observations should be mentioned. Ground-based MW instruments operate with very high temporal resolution (1-2 second), continuously for very long periods of time, in unattended mode, independently of solar illumination and nearly at all weather conditions. The evident advantage of satellite observations is their global scale, however the MW satellite sensors deliver the information only over water areas since the emissivity of the land surface is highly variable. The superiority of the SEVIRI instrument working in visible–near-infrared range is the possibility to make observations over water areas and land surface as well, however only when the atmosphere is illuminated by Sun since the instrument measures the reflected solar radiation."

The referee puts the question about the detection limits of the methods. Since the ground-based and satellite sensors have extremely different spatial resolution and therefore probe considerably different portions of atmospheric air, this question can not have a definite answer and requires an analysis of different specific atmospheric situations. This is the reason why we could not address this point in the revised version.

In the revised version we present a two dimensional histogram with number of occurrence colour scale, which, according to referee's opinion, best illustrates the agreement of both data sets. This histogram is given in Fig. 9 (the numeration of figures of the revised version) and the text with the corresponding description is given in Section 5.1:

[Figure]

Figure 9: Comparison of the HATPRO and SEVIRI "instantaneous" measurements by means of two-dimensional histogram with number of occurrence colour scale. Upper panel: extra high LWP values are shown, lower panel: only LWP<0.4 kg m$^{-2}$ are shown.

"We begin our analysis making a comparison of the instantaneous HATPRO and SEVIRI measurements of LWP by means of a two-dimensional histogram with the number of occurrence colour scale that is displayed in Fig. 9. This plot gives an impression about the overall agreement of measurements disregarding seasonal features. First of all, attention should be paid to the presence of a noticeable number of very high LWP values

detected by the SEVIRI instrument and reaching 2.3 kg m$^{-2}$. However, the number of occurrence of these measurements is very small if compared to the number of occurrence of the small values. The two-dimensional histogram for LWP<0.4 kg m-2 shown in the lower panel of Fig. 9 demonstrates that the largest number of occurrence is observed for small LWP not exceeding 0.03 kg m$^{-2}$. The agreement between HATPRO and SEVIRI data for these values is good. For higher values, the agreement is not evident. This fact is not surprising since the agreement between instantaneous measurements is influenced by mistime, misdistance, weather conditions, type of cloudiness and the parameters of time averaging of the HATPRO data."

**2) The LWP difference between land and sea for is strong and is one of the most interesting points of the paper. The paper takes it for granted that this is real but there needs to be a discussion/investigation whether this might be caused by a shortcoming of the SEVIRI, e.g. maybe due to the difference in surface albedo between land and sea. Furthermore, if it is true, a physical explanation for the LWP gradient needs to be provided. A potential explanation is the frequent presence of a high pressure system over the Baltic Sea and the associated subsidence which causes adiabatic warming and low cloudiness. With this explanation it might be better to separate the LWP time series into weather type situations rather than warm & humid (WH) and cold & dry (CD.**

This comment of the referee can be divided in two parts. The first part concerns the approval of the land-sea LWP difference. In order to demonstrate that the difference is real, we refer to the results of the cloud amount study in the Scandinavian region by Karlsson (2003):

Karlsson, K.: A 10 Year Cloud Climatology Over Scandinavia Derived From NOAA Advanced Very High Resolution Radiometer Imagery, Int. J. Climatol., 23, 1023–1044, DOI: 10.1002/joc.916, 2003.

We have included the following text which describes these results in Section 3:

"It should be noted that the land-sea differences of cloud characteristics in Northern Europe have been detected earlier. Karlsson (2003) compiled regional cloud climatologies covering the Scandinavian region on the basis of processing data from the NOAA Advanced Very High Resolution Radiometer (AVHRR) instrument for the period 1991–2000. Considerable local-scale variation of cloud amounts was found in the region. During the spring and summer seasons, as a contrast to winter and autumn conditions, much less cloudiness has been found over seawater and major lakes. It has been suggested that the cold sea surface temperatures in the Baltic Sea (especially in spring and early summer due to inflow of cold fresh water from melting snow) lead to a considerable stabilization of near-surface layer of the troposphere. This explanation agrees well with the fact detected in our study: the land-sea gradient in the mean LWP values for the CD period. is noticeably lower than for the WH period."

The second part of the comment concerns the explanation of the land-sea difference and the use of such explanation as the basis for the criterion to separate the data into subsets. One explanation has already been suggested by Karlsson (2003) and our results agree well with this explanation. The explanation proposed by the esteemed referee could be one of the reasons, however not the main one since the land-sea borders on the LWP maps are well-defined, and the effect takes place also in the vicinity of Ladoga lake where the boarders are well-defined too. Concerning the referee's proposal to separate the LWP time series into weather type situations:

1) We agree that analyzing the weather situations is generally a good idea.

2) To our opinion the weather analysis is very advantageous when considering the specific cases, as we did when we considered the cases of bad agreement between satellite and ground-based data and revealed the presence of rain events in the investigated region.

3) The separation of data according to weather type could be to a great extent arbitrary and ambiguous since the weather is a multi-parameter phenomenon.

**3) LWP statistics: LWP is highly variable in time and space and this variability strongly depends on the cloud type, i.e. is strongest for convective boundary layer clouds. Therefore, it is difficult to make solid statistics even if a two year data set is considered. By spitting the data further into individual months and climatic conditions the distributions become rather erratic and should not be overinterpreted. Smooth distributions require rather long time series (see Caddedu et al., 2013, Kniffka et al., 2014). Therefore, I recommend to just separate into the warm/humid and cold/dry regime or seasons at the most.**

We agree with this comment and as far as we understand, it refers to our analysis of monthly frequency distributions and monthly diurnal cycles. Addressing this comment, we changed the plots in Fig. 11 and Fig. 12. Fig. 11 contains now the frequency distributions for seasonal periods (WH and CD) and Fig. 12 displays the monthly distributions only for six months with the largest number of instantaneous measurements. The analysis of monthly distributions is necessary since the distributions for the WH and CD periods differ noticeably. We edited accordingly Section 5. Subsections have been reorganized. Subsection 5.1 now reads:

"5.1 Seasonal features

[revised manuscript text omitted]

Concluding the reply to the comment No3, we would like to stress that the analysis of diurnal cycle is not possible for large seasonal time intervals since the illumination conditions differ considerably from month to month. Therefore, we kept the analysis of diurnal cycle unchanged.

**4) In respect to statistics the comparison with reanalysis is also difficult as only one instantaneous value every 3 hours is provided and thus only 8 per day and is not comparable with the better sampling of SEVIRI and MWR measurements. Thus it is the question whether the interannual variability shown add the end study is due to sampling or real and would require testing of the statistical significance. While I find it very important to make the point of high interannual variability I would remove the reanalysis aspect from the study as the data are not comparable in terms of spatial (80 vs 10 km) and temporal (8 to about 50) scales even with the coarser SEVIRI LWP data and also represent a mixed land & sea pixel.**

We agree with the statement that due to differences in resolution, the comparison of the reanalysis LWP data with the data from SEVIRI and HATPRO is problematic. Nevertheless, the referee found important to emphasize the interannual variability. Therefore we decided to remove the comparison aspect but to keep the discussion of interannual variability and to place it and relevant plots in the short Appendix 2 "The diurnal cycle interannual variability as derived from reanalysis data". We also included the following text in the end of Section 6:

"Concluding this section, we would like to emphasize the importance of taking into account the interannual variability of diurnal cycles. Our estimations of the interannual variability were based on the reanalysis data and are presented in Appendix 2. It has been shown that the average diurnal cycles calculated for the period of our study (2013-2014) noticeably differ from cycles obtained for the longer period 2003-2012. Since the

temporal and spatial resolutions of the reanalysis data are considerably coarser than of the SEVIRI and HATPRO data, the direct comparison of diurnal cycles are not possible."

**5) The paper contains many plots and many could be eliminated. Why are s lengthy by showing both median and mean LWP. What is the benefit? The LWP distribution is strongly skewed towards low values and thus the median LWP values are typically lower than the accuracy. I would suggest to keep only the mean. If you would like to show the median then you could put it into an appendix. M. P. Cadeddu, , J. C. Liljegren„ and D. D. Turner, 2013: The Atmospheric radiation measurement (ARM) program network of microwave radiometers: instrumentation, data, and retrievals. https://www.atmos-meas-tech.net/6/2359/2013/amt-6-2359- 2013.pdf**

We cannot agree with the esteemed referee that the paper is overloaded with plots. The referee presented the example with LWP maps where both mean and median values are shown. To our opinion, the median value is the best for showing the "typical" quantity and this fact is very important for description of cloudy atmosphere with the presence of a large number of clear situations. However we followed the advice of the referee, we reorganized the plots and put the LWP maps with median values in Appendix 1.
* * *
**MINOR COMMENTS:**

**L13: Provide also the relative error as the mean LWP is rather low and thus errors should often be in the order of 100 %.**

We do not agree with this comment because the cloud-free situations have been included in the analysis but relative error has no sense for such situations and also due to the presence of such situations the mean LWP is low.

**L22: You just report on what you did but what was the result? Please extent.**

The new version reads:

"On the basis of reanalysis data, it has been shown that the LWP diurnal cycles are characterized by a considerable interannual variability."

**L40: There are certainly many more studies on (sub)arctic clouds than the one by Garett and Zhao, the point to make here is that the measurement network is rather coarse in that region.**

The new version reads:

"Several ground-based microwave radiometers are permanently functioning at Northern latitudes as the elements of the MWRnet - An International Network of Ground-based Microwave Radiometers (http://cetemps.aquila.infn.it/mwrnet/main_files/MWRnetmap.html), however the measurement network is rather coarse in that region."

**L52: Here, you need to emphasize on the importance of LWP as essential climate variable which is difficult to assess due to its high spatio-temporal variability, cf. Van Meijgaard and Crewell**

**(2005) for the difficulties to compare LWP with models of different grid size.**

The importance of LWP as essential climate variable has been addressed already in the "Major comments" section above. We added the reference to (Meijgaard and Crewell, 2005):

"Special measurements campaigns with microwave radiometers have been carried out in Europe with the focus on liquid water path and the difficulties have been demonstrated to compare the measured LWP with models of different grid size (Meijgaard and Crewell, 2005)."

**L54: You need to explain the two satellite measurement principles for LWP from satellite and provide their limitations and uncertainties. 1) VIS/NIR observations only possible during daylight (not mentioned in manuscript) but available from geostationary satellite. This method needs to make assumptions on the vertical structure of the clouds as only the top can be sensed. 2) Microwave imagers on polar orbiters measure the emission signal which is only possible over the radiatively cold ocean - here you can cite Elsaesser et al. (2017) for the climatology and Greenwald et al. (2018) who investigates the uncertainty albeit by taking VIS/NIR as truth. Note, that due to the large footprints and the differences in emissivity between land and ocean no information for coastal pixels is available.**

These points have already been addressed in the in the "Major comments" section above. We added the recommended references:

"The climatology of LWP obtained from satellite observations have been presented by Elsaesser et al. (2017). The importance of combining visible-infrared imager data and passive microwave LWP observations for estimating uncertainties and improving the accuracy of these observations has been demonstrated by Greenwald et al., 2018."

**L55: I don't think it is necessary to list the satellite instrument names - if you do you need to provide the explanations for all acronyms.**

The satellite instrument names have been removed.

**L69: The uncertainty of ground-based MWR for LWP needs to be given**

It is given in Section 2.1 of the revised version as an answer to the major comment.

**L87: Also the size of the footprint is larger than at the other locations**

The text has been inserted:

"Also the size of the footprint is larger than at the other locations."

**L92: "Since the LWP values can be essentially different over.." has this been shown in the literature or is this your result?**

These point has already been addressed in the in the "Major comments" section above. Here we only added the reference to the paper by Karlsson (2003):

"Since the LWP values can be essentially different over land and sea surfaces (Karlsson, 2003), and taking into account…"

**L95: Not give only one of the main goals but provide them all, ...for example the investigation of reanalysis quality...**

In the revised version, the end of Section 1 is the following:

"So, the main goals of the present study were to identify the problems of the comparison of HATPRO and SEVIRI measurements of LWP at high latitudes over the complex terrain which includes land and water areas, to analyse the frequency distributions and diurnal cycles derived from measurements of the two instruments and to assess systematic and unsystematic discrepancies between the satellite and ground-based data sets."

**L99: Why not cite Rose et al, 2005 at this stage - it is cited on 160**

The reference has been added:

"The 14-channel microwave radiometer RPG-HATPRO (generation 3) developed for the retrieval of temperature and humidity profiles in the troposphere along with LWP and integrated water vapour (Rose et al., 2005) has been routinely functioning…"

**L129: 11km? The size at St. Petersberg needs to be mentioned**

Mentioned:

"In the vicinity of St.Petersburg the ground pixel size is about 7 km."

**L144: It is not clear to me why only days which are completely free of rain are included in the comparison? This eliminates many data which are needed as the high variability of LWP makes statistical analysis difficult. If days with precipitation are excluded the mean diurnal cycle derived later on is only the mean of a subsample and not overall. At least this needs to be made clear maybe by giving it another name.**

For clarity, the following text has been added:

"The reason to completely exclude from consideration the days with rains is the following. Not only during a rain event but also for a rather long period of time after it, the data provided by HATPRO are erroneous since the radio dome of the instrument is wet. The duration of this after-rain period for St.Petersburg site has been estimated in the study by Kostsov et al. (2017) as 4-6 hours. Moreover, it has been demonstrated in the mentioned study that the situations are possible, when the measurements are erroneous even before the rain event, when the rain sensor is not yet detecting the rain signal. It is evident that even one rain event during a day results in the considerable loss of data."

We also added the following text in the beginning of Section 6 "LWP diurnal cycle analysis":

"It is necessay to remind that the initial datasets consist of rain free days only, therefore the analysed diurnal cycles do not present the overall estimate but only the subset of purely liquid clouds during rain-free days."

**L146: How is a gap defined - a single 1 sec measurement?**

The text now reads:

"The measurement process must not have had gaps which are defined as 15 min or more period without measurements."

**L147: You speak about convergence - is the quality flag from the physical retrieval?**

The text now reads:

"The quality flag of MW measurements must have been zero for all retrievals that means the successful convergence of the iteration process of physical retrieval for every single measurement."

**L160: In general, I think you need to explain why this is important: The retrieval algorithms are typically non-linear. Therefore, the retrieval needs to be made on high resolution brightness temperature data and subsequently then the LWP can be averaged but not vice versa. Note, that since the paper by Rose et al is already more than 10 years ago and since when resolutions of NWP have increased. It should be mentioned that situations can be very different and in particular convective boundary layer clouds have high variability.**

We followed this recommendation and the following text has been included in Section 2.3:

"It is important because the retrieval algorithms are typically non-linear and therefore the retrieval needs to be made on high temporal resolution brightness temperature data for subsequent averaging of the results but not vice versa. Also, it should be mentioned that situations can be very different and in particular convective boundary layer clouds have high variability."

**L165: Please also write "stable" as this has nothing to do with the thermal stability but more with constant conditions. .**

The text has been changed, now it reads:

"The obtained results have shown that even for constant atmospheric conditions the sampling interval should not be greater than 100-200 s in order that maximum information could be extracted form MW measurements. Though this conclusion had more theoretical value than a practical one, for the present study we have chosen two original MW datasets that differ by the sampling interval: 120 s and 10 s."

**L170: For a comprehensive discussion on how to compare LWP from ground-based observations and with spatial estimates from NWP models of different resolutions see Van Meijgaard and Crewell., 2005.**

The reference has been added:

"The comprehensive discussion of the aggregation of the ground-based LWP data to coarser time scales has been presented by Meijgaard and Crewell (2005) who considered the comparison of ground-based LWP observations with the estimates from NWP models."

**L180: Why boxcar? For simplicity?**

Yes, for simplicity. We mentioned it in the new version:

"…however the weighting function has been taken not Gaussian but a boxcar for simplicity."

**L190: As Kostov et al. (2016) is not an open access paper the values for distinguishing WH and CD should be provided here. There is no surprise that WH is more frequent as SEVIRI needs light.**

We inserted the definition of time intervals corresponding to WH and CD periods:

"The corresponding time intervals are: 1 May – 30 November and 1 December – 30 April."

**L200: How good is the the geolocation of SEVIRI?**

The following text has been inserted in the beginning of Section 3:

"The accuracy of SEVIRI's geolocation depends on the actual satellite on which the instrument is mounted and amounts approximately to 1.32 km in north-south direction and 0.15 km in east-west direction as stated in the document on MSG level 1.5 image data description (EUMETSAT, 2017) plus an additional error of 1.5 km in both directions in the data prior to 2017 because of an undetected pixel offset."

**L203: especially as liquid clouds are at low heights.**

The text has been edited correspondingly:

"Obviously, this influence is not significant for homogeneous cloud fields, and for clouds at low heights."

**L227: The statement is not correct - I still see the gradient.**

We have the feeling that there is a kind of misunderstanding. We checked the former Fig. 5, right panel, map of median values for CD period. The gradient does not exist, all input data are zeros, and this fact is explicitly stressed in the text. In the revised version this plot is in Appendix 1, and the absence of the gradient is emphasized in a special note.

**L240: Different sampling and averaging times are chosen which provides some information on the error introduced by representativeness of the measurements. However, the optimal combination should depend on the actual weather situation - which should be discussed. Could the spatial distribution of SEVIRI or NWP might provide information to optimze this on an individual base.**

We do not agree with this comment. The dependence of the averaging time on the weather conditions is an obvious conclusion and needs no discussion. Besides, working out the criteria for choosing the averaging time value on the basis of current weather conditions was not the task of the study.

**L257: The agreement in daily mean LWP between SEVIRI and HATPRO in Fig. 7 is claimed "very good". I don't think so as many days especially in the beginning have zero LWP in SEVIRI and relevant**

**values for HATPRO. Try to make a more quantitative statement here. You could also make a table for the daily means similar to table 4**

Fig. 7 shows not mean but median values. The median values represent "typical" quantities and zero LWP indicates the fact that cloud-free conditions were prevailing. That does not mean that clouds were absent during a whole day of measurements, Therefore the agreement of SEVIRI and HATPRO values which are close to zero is a good demonstration of the overall agreement between measurements. We think that there is no need for a quantitative statement and an additional table here.

**L262: Is this for instantaneous or daily LWP?**

This is for daily median. We changed the text and the Table caption accordingly.

**L290: Section 2.2. needs to mention that SEVIRI retrievals fail in the case of strong vertical gradients and especially during rain. Why not use standard deviation of 3x3 SEVIRI pixels to identify (and eliminate) inhomogeneous situations?**

The proposal of the referee is good, we shall take it into account in the subsequent studies, but in the present study, when selecting the data for comparison, we deliberately used the criteria based only on the HATPRO measurements as described in Section 2.2. Nevertheless, as recommended by the referee, we mention that SEVIRI measurements fail in case of rains:

"It should be mentioned that SEVIRI retrievals fail in cases of strong vertical LWP gradients and especially during rain events."

**L329 and Table 4: Why has this analysis not been done over the whole time period? It is important to do for generalization and it is a good transition to the next section.**

As it has been stated in the Introduction, our goal was to identify the problems of the comparison of HATPRO and SEVIRI measurements of LWP at high latitudes. We plan to continue our investigations with special attention to the parallax effect and land-sea gradients. However we would not like to expand the current study.

**L332: The title of the section should be changed to something like "Statistical LWP assessment". In this section I would have expected you - after looking at the mariginal distributions - to show the joint distribution which I am strongly missing here. This provides a much more direct view on the systematic and unsystematic components.**

The title of the Section has been changed to "Statistical LWP assessment". The joint distribution with colour scale has been placed in this section.

**In this sense L349: The name of section "5.1 Instrument differences" is a bit misleading. With this title I would have expected a discussion on the different sensitivities of the instruments. I think no subsection headings are needed here. I recommend the authors to look at Tian et al. 2016 https://journals.ametsoc.org/doi/pdf/10.1175/MWR-D-15-0087.1 for a better error model.**

We would like to keep the subsections, but following the recommendation of the referee, we changed the title of the subsection to "Analysis of discrepancies". We are grateful to the referee for the recommended very interesting paper to look at. However we prefer to keep the systematic and nonsystematic error analysis unchanged.

**L335: If you say "low compared to" but don't give any values this becomes difficult for the reader. It is also not possible to compare it with the full disk as many different climate zones are - better compare with other mid-latitude sites, cf. Cloudnet sites.. Her you should integrate the section 5.2 which somehow does not fit to the end. There a range for the distribution 0.2 - 0.6 kkgm-2 is given. This is better quantified by an interquartile or interpercentile range. Would be also good to give the numbers for the mean in both regimes. I**

We described the data in comparison to the average values of the whole disc in order to give a general characterisation of the measured liquid water paths. It was meant to show how the data fit into the worldwide range of the SEVIRI disc's LWP retrievals prior to the start of the more detailed analysis. The aim was NOT to compare to equal climate zones or mid-latitude sites, which was also not the scope of the article by Kniffka et al. (2014). We will therefore keep it in this way and add the average values for the two analysed periods. Please note, that the mentioned paragraph has already changed significantly, therefore we reprint here only a small part:

"The distributions do not fall directly into one of the four categories in Kniffka et al. (2014), where all cloud types were characterised with mono-modal distributions, however they do resemble the low clouds category the most. The average all-disc values range from 0.0672 to 0.0862 kg m$^{-2}$ (depending on the season) while the average HATPRO (SEVIRI) LWPs amount to 0.0182 (0.0274) and 0.0243 (0.0310) kg m$^{-2}$ for the cold, dry and warm, humid period. The climate of St.Petersburg is maritime where low stratiform clouds occur most frequently. Thicker, presumably convective clouds with LWP $> 0.1$ kg m$^{-2}$ form the secondary maximum in the distributions and occur in both periods (in the end of the CD period and in the beginning of the WH period)."

**L340: I don't think that you can generalize the bimodal distribution for all February and September months with only two years and so much fluctuation in LWP. This is just due to certain weather patterns.**

As it has been written above in the "Major comments" section, we removed months with low number of measurements from the analysis, February was one of them. However, it was possible to keep September. We would like to stress that we do not make generalizations when we analyse the statistical characteristics.

**L340: Talking about 17% creates an impression of high accuracy - albeit the distributions are highly uncertain as they are determined by very view events and bimodality just comes by chance. You should provide the total number of samples for each distribution and as I suggest in the beginning just concentrate on the broader picture.**

This point has already been addressed in the "Major comments" section, the number of samples has been given. The bimodality does not come by chance, it is clearly visible in spring and early summer months.

> **L345: Kniffka et al have a much larger and solid data base so that the comparison of the distributions is not fair here. Especially the bimodality (secondary maximum) is most likely just by chance and would not exist if a decade of data would be used – this could be checked by SEVIRI.**

We agree that bimodality may not exist if a decade of data is taken. However, the point is that for the considered time interval (2 years) this type of distribution has been detected for several months in a row. This is the fact, and we would like to stress once again that we do not make any generalizations.

> **L355: Some background needs to be given on the method and an assessment of the "uncertainty".**

To our opinion, the given reference (Anand et al., 1991) is sufficient.

> **L361: I would have expected the highest error in summer as the high variability of convective clouds reduces the representativeness of the HATPRO measurements for the SEVIRI footprint and complicates the comparison in contrast to situations with more stratiform conditions typically more frequent in winter. cf. Slobodda, J., A. Hunerbein, R. Lindstrot, R. Preusker, K. Ebell, and J. Fischer, 2015: Multichannel analysis of correlation length of SEVIRI images around ground-based cloud observatories to determine their representativeness, Atmos. Meas. Tech. , 8, 567-578, doi:10.5194/amt-8-567-2015.**

In the revised version we mentioned the better viewing conditions for SEVIRI only and added the speculation about representativeness of the HATPRO measurements:

> "However, the error could increase in summer due to high variability of convective clouds which reduces the representativeness of the HATPRO measurements for the SEVIRI pixel and complicates the comparison in contrast to situations with more stratiform conditions typically more frequent in winter. The detailed discussion of the problem of representativeness can be found in the paper by Slobodda et al. (2015)."

> **L360: As the authors also mention later on clouds with LWP of >0.4 kgm-2 are likely to precipitate and therefore I would suggest the authors to eliminate all SEVIRI retrievals above a certain threshold as the retrieval is not built for rain. Furthermore, it is only fair as rainy HATPRO measurements are eliminated as well.**

We eliminated the cases with LWP>0.4 kg m-2 when we plotted seasonal and monthly frequency LWP distributions but we deliberately kept all positive values including high ones for error analysis.

> **L458: Why not mention the corresponding advection velocity?**

Mentioned in the revised version:

"There is no influence either of the sampling interval (10 s or 120 s) or of the averaging period (20 min or 60 min) of the original HATPRO data on the results of the HATPRO-SEVIRI data comparisons. (The given values of the averaging period correspond to the values of advection velocity of about 6 m s-1 and 2 m s-1.)"

**L464: What is the accuracy of the 3.1 km estimate? I would simply say around 3 km.**

Corrected.

**L368: and the large footprint at this high altitude.**

Correspondent editing has been made:

"The second site-specific feature is the high latitude location of the radiometer site and the large pixel size at this high latitude."

**L479: Why do you know that it is log-normal? Did you test it? For this you probably need many more data so I would be more careful with the statement. Similar with the bimodality which is certainly due to poor sampling of different weather types and not a climate phenomenon.**

We did not perform rigorous testing for lognormality and bimodality but made qualitative fitting. So, we are confident in our statements. Also, we would like to note that we do not present our results as a climate phenomenon.

**Table 3: You also need to mention the number of samples, the mean value (so one can assess the relative error), bias corrected RMS and the correlation. Which SEVIRI pixel is taken?**

We have corrected the text and clarified what values are compared (daily median):

"The estimates of the bias and rms difference between the **daily median** LWP values derived from satellite and ground based observations are given in Table 3."

So there is no need any more to present the number of samples since this information is given in Table 2.

The mean values are now presented in the text:

"The daily median values averaged over the datasets constitute 0.017 kg m-2 and 0.02 kg m-2 for WH and CD datasets correspondingly."

Since bias is very small, presenting the bias corrected RMS difference would not be very helpful.

We modified the table and presented the correlation coefficients between different datasets. Correspondingly, the following text has been added:

"It should be emphasized that the correlation coefficients for the WH season are considerably larger than for the CD season."

In order to clarify what SEVIRI pixel is used, we added the following remark in Section 2.3:

"It should be noted that all comparisons have been made for SEVIRI ground pixel which is the nearest to the radiometer site. In case other pixels are considered, it will be mentioned explicitly."

**Table 4: Not only the four cases but also the whole time period and WH and CD should be shown.**

This is a case study. Our intention was to demonstrate the parallax effect only for separate cases. For the complete dataset this effect can be masked.

**Figure 1: It is a bit irritating that pixel 242 is discussed but not shown.**

This is not true. The centre of pixel 242 is clearly visible and labelled in the upper right corner of the right panel in Fig. 1.

**Figure 3: The contouring provokes more interpretation than possible from only 3x3 points. Please plot them as blocks. Maybe you can even integrate them into the lowere left corner of fig. 2 as a zoom in. Same holds for all figures with 3x3 points, e.g. 4 and 5. Then you can also show WH and CD together.**

We do not agree that the interpolation of 3x3 points for contouring can lead to overinterpretation of plots. The lines are very smooth and no high resolution artefacts are present. Moreover, the interpolated plots better demonstrate the correspondence of the LWP map to geographical map. Therefore we kept all 3x3 point maps unchanged. Since the maps with median LWP have been moved to appendix, in the revised version of the paper the WH and CD maps are shown side by side in reorganised plots.

**Figure 6: You should give the number of samples, bias, rms, bias corrected RMS and correlation in a table similar to Table 3.**

We do not agree with this comment. The purpose of this Figure is to demonstrate that the influence of sampling interval can be neglected if compared to the influence of the averaging period. And it is clearly seen from the Figure. Presenting the numbers would not add any substantial information.

**Figure13: Which averaging do you use here? I am again concerned with the statistics. Either leave this out as monthly data have poor statistics (give number) or if you want to do it keep it then use the full Taylor diagram. https://climatedataguide.ucar.edu/climatedata-tools-and-analysis/taylor-diagramspart van Meijgaard, E., and S. Crewell, 2005: Comparison of model predicted liquid water path with ground-based measurements during CLIWA-NET. Atmos. Res., Special issue: CLIWA-NET: Observation and Modelling of LiquidWater Clouds, 75(3), 201 - 226, doi:10.1016/j.atmosres.2004.12.006.**

First of all, the statistics is not so poor. Most of the months include 200-500 data samples, August and October have about 150 samples each and there is only one month (February) with the lowest number 45. Second, we do not make any climatological conclusions and only analyse the character of discrepancies for different conditions. For such a task the number of samples is sufficient. We are grateful to the esteemed referee for the very useful recommendation and

reference but as far as using the Taylor diagram instead of the approach by Anand et al. (1991) is concerned, we think this is a matter of choice of the authors.
* * *
**Grammar, typos and reformulations**

**L10 - spell out SEVIRI and mention geostationary**

We tried to keep the abstract short and clear. Therefore no peculiarities of both instruments are given in the abstract. All necessary information about the instruments is presented in the main text of the manuscript.

**L4: "have shown considerable differences" in you study (cf Fig.4) or in literature. Always make clear what are your results and what is known from the literature**

We reformulated the sentence:

"The radiometer measurement site is located very close to the shore of the Gulf of Finland, and our study has revealed considerable differences between the LWP values obtained by SEVIRI over land and over water areas in the region under investigation."

**L25: - this is not the only interest, I suggest: "There is increasing interest in the sub-Arctic region due to the so-called "arctic amplification" effect that .."**

Now the sentence reads:

"The interest of studying the sub-Arctic atmosphere is enhanced due to the so-called "arctic amplification" effect."

**L28: "The large seasonal and interannual variation in low- and high-pressure systems and associated environmental variability due the location of the Baltic Sea between the North Atlantic and Eurasian air masses makes North Europe especially important to study.."**

Corrected, now the sentence reads:

"The large seasonal and interannual variation in low- and high-pressure systems and associated environmental variability due the location of the Baltic Sea between the North Atlantic and Eurasian air masses makes Northern Europe especially important to study atmospheric processes (Eriksson et al, 2007)."

**L57: "This study exploits the ... operating at..."**

Corrected.

**L65 Cloudnet not explained**

We added the internet site reference for CloudNET:

"Roebeling et al. (2008a) determined the accuracy and precision of LWP retrievals from SEVIRI on board Meteosat-8 using 1 year of LWP retrievals from microwave radiometer measurements of two CloudNET (http://www.cloud-net.org/) stations located in the United Kingdom (Chilbolton) and France (Palaiseau)."

**L112:"in every detail" -> in detail**

Corrected.

**L125: Order of Dee and Dankers**

Corrected.

**L133: Channel "at" 1.6**

Corrected.

**L129: One sentence does not make a paragraph.**

Corrected.

**L171: Suggestion: Roebeling et al. (2008a) assumed Taylor's frozen turbulence hypothesis to relate an assumed wind speed of 10 m/s SEVIRI field of view (4 x 7 km2 resulting in a 20 min averaging period which Roebeling et al (2008b) extended to 30 min.**

The wind speed assumed in the present study has been given. (See one of the comments above).

**L182: The term synchonization does not fit so well - better use collocation or coincidences.**

We would like to keep "synchronization".

**L184: CPH - all acronyms need to be explained period assuming the wind speed about 10 m s-1 and the).**

The acronym CPH is explained in the end of Section 2.2.

**L185: I think the argument goes into the wrong direction. The MWR gives you LWP in all situations but SEVIRI can not do so in case there is ice. Better write "Since SEVIRI cannot retrieve LWP..."**

The sentence has been reformulated:

"Only liquid phase clouds have been considered, therefore all SEVIRI measurements with CPH=2 have been excluded from further analysis and from synchronization with HATPRO results."

**L201: Greuell and Roebeling (2009) studied the influence of the parallax effect (the horizontal displacement of a cloud viewed by SEVIRI due to its elevated height) on the comparison of SEVIRI and ground-based microwave LWP:**

We used the definition of the parallax effect that was given by Greuell and Roebeling (2009), therefore we kept the sentence unchanged.

**L285: rain**

Corrected.

**L354: "The coloured circles...." The figure caption does not need to be repeated.**

The sentence

"The radii of the circles correspond to the monthly averaged RMSE values."

has been moved to figure caption.

**L369: ..that the possible,,**

The sentence now reads:

"On the basis of this information we suggest that the possible supercooled clouds with simultaneously very high effective radii can be the indication of the presence of erroneous retrieval results."

---

## Author Comment (AC4) · 28 Aug 2018

**The reply to the anonymous referee #2 (RC4)**

We are thankful to the referee for the careful reading of our manuscript and for the valuable comments. We appreciate all the comments; we took them into account while preparing the revised version of the manuscript.

Below, the actual comments of the referee are given in **`bold courier font and blue colour`**. The text added to the revised version of the manuscript is marked by red colour.

> **`Section 2.1: This section should contain information about the calibration procedure of the HATPRO. How often where calibrations done? Is there any correlation between time since last calibration and quality of agreement between HATPRO and SEVIRI?`**

We added the information about calibrations in the end of Section 2.1:

> "There were 13 calibrations of the instrument during this period of time including 7 absolute calibrations with liquid nitrogen and 6 sky-tipping calibrations. The interval between absolute calibrations varied from 2 to 4 months."

We did not estimate the correlation between time since last calibration and the quality of agreement between HATPRO and SEVIRI because the HATPRO measurements of LWP for the whole period of time under consideration have already been validated. And this fact has been mentioned in the original text: "…(3) the measurement data have already been validated and analysed for this time period (Kostsov et al., 2018)."

> **`Line 159: Is sampling equal to averaging? If not, what is meant by 'sampling interval'? Does it mean taking just a single 1-s value every X seconds? If yes, why is this done? To safe disk space? To get into better agreement to the temporal resolution of SEVIRI?`**

Sampling interval is the time interval between instantaneous measurements by HATPRO. Sampling period is the time of averaging of the signal coming from the atmosphere; it may be also called the integration time. In order to clarify this point we changed the text in the middle of Section 2.3. Now the text reads:

> "The sampling interval (the interval between instantaneous measurements) of routinely performed ground-based MW observations is about 1-2 s since the sampling period (the integration time of the incoming atmospheric signal) is equal to 1 s."

We have the feeling that the confusion was caused by the sentence:

"It has been noted by Rose et al. (2005) that the integration time (**or** the sampling interval) should not be greater than 20 s…"

Now this sentence reads:

> "It has been noted by Rose et al. (2005) that the integration time (**and also** the sampling interval) should not be greater than 20 s…"

In order to explain why the problem of variable sampling interval is important, we added the following text:

> "However, there are situations when keeping the sampling interval as small as possible is problematic. If an instrument is functioning in the mode of azimuth scanning or zenith scanning, some time is needed to change

pointing. Also, an instrument can be set to make the mixed mode observations. In this case the interval between measurements made in a certain mode can be rather large."

**Lines 195-200: Suddenly the 'Petergof' site is introduced, but where does it come from? Is this the location of the MW? If yes, it would fit to introduce Petergof in line 195: "...measurement site of Petergof."**

Yes, the radiometer is located near the railway station Petergof. In order to avoid confusion, we replaced "Petergof" by "St.Petersburg" in the revised text.

**Lines 262-263: Is there any possible explanation for the found extremely large discrepancy? Such a statement leaves a very curious reader...**

We pointed that the explanation will be given below:

"Since there was only one day with extremely large discrepancy between the results (day No 52), we excluded this day from the calculations (this is 14 May 2014 and the reasons for large discrepancies are discussed below)."

**Line268ff: Which Seviri pixel was used for the comparison shown in Figure 8? Also at other positions in the text it was not always clear whether the shown results of SEVIRI are an average of the whole domain or only of a single pixel. If there was a general procedure applied, the authors should present it in Section 2.3.**

We added the following remark in Section 2.3:

"It should be noted that all comparisons have been made for SEVIRI ground pixel which is the nearest to the radiometer site. In case other pixels are considered, it will be mentioned explicitly."

**Line 312: Is 'not less' the same as 'at least'? If so, at least would lead to less confusion while reading.**

In order to avoid confusion the sentence now reads:

"Based on these values and accounting for the higher latitude of the St.Petersburg measurement site, we can expect the parallax effect for the St.Petersburg measurement site to be about 3 km or more in terms of the displacement to the North direction."

**Line 483: What determines 'systematic' and 'unsystematic' discrepancies?**

To our opinion, the answer to this question asked by the referee can be placed in Section 5.2 "Analysis of discrepancies" before the discussion of unrealistically high LWP values produced by SEVIRI:

"The sources of systematic and unsystematic discrepancies are multiple. They can be related to the retrieval algorithms, parameters of time-averaging of HATPRO data, viewing conditions of SEVIRI, and also to weather conditions, type, height, spatial and temporal evolution of clouds, and the magnitude of parallax effect. The analysis of the details of retrieval algorithms is beyond the scope of the present study. The variation of the averaging period was shown to have minor influence on the results of comparison. Therefore, while making the analysis we focused on weather and cloudiness conditions provided by the SEVIRI observations simultaneously with LWP data."

**Lines 487-489, or Line 510: Can the authors present ideas, which parameters should be investigated to identify the reasons for the occasionally observed discrepancies? Would the analysis of rain radar data help? Or can it be attributed to strong small-scale fluctuations of LWP?**

We presented one of the ideas in the end of item 5) in Section 7:

"To our opinion, in order to further analyse the reasons of these systematic differences, it would be useful to combine the HATPRO and SEVIRI data with collocated LWP data produced by the AVHRR instrument. Though the LWP measurement over St.Petersburg site is made by AVHRR only once per day, the size of the ground pixel of AVHRR is smaller than of SEVIRI and this fact would be very helpful."

**Line 165: situation .... situations**

Corrected. Now the text reads:

"…for constant atmospheric conditions…"

**Line 166: form ... from**

Corrected.

**Line 215: superscript m^-2**

This sentence has been rewritten.

**Line 231: "...not one, but two SEVIRI pixels..."**

Corrected.

**Line 239: "...20 min and 60 min, respectively."**

Corrected.

**Line 346: mono-model ... mono-modal**

Corrected.

**Lines 406-407: "...for the summer months..."**

Corrected.

**...there are much more typos present in the manuscript...**

We checked the text and tried to correct all typos.

**Figures 2-5: The same scale should be used for all figures. At the moment, each figure shows a different range of values, making a visual comparison impossible. Also, it would be of interest, how many data points are contained in each map. Could the authors provide a map of the number of data points per pixel?**

We do not agree with the referee that the same scale should be used for all maps. The primary goal was to demonstrate the land-sea LWP gradient. We specially chose the scale and colours in order to reach this goal and to emphasize that the gradient area corresponds to the coastline. Therefore we kept the Figures unchanged in the revised version of the manuscript.

We indicated the number of data points per pixel not in the plot but in the text in Section 3:

"Fig. 2 presents the maps of the mean LWP values calculated for the large and small terrains and for the whole considered 2-year period of observations (about 20000 data points per pixel)."

and

"In order to assess whether this gradient can influence the results of the comparison of the SEVIRI and the HATPRO data, we plotted similar maps only for the data selected for comparison and considered the WH and the CD periods separately (about 4000 and 2000 data points per pixel respectively), see Fig. 3."

**Figure 14: Please show a legend.**

We do not understand this comment. Figure 14 and Figure 15 are similar and both have the description of lines in the figure caption. Since the plots are simple with only two types of lines, the legend is not necessary.

---

## Author Response (AR1)

[revised manuscript text omitted]

------------------------------Разрыв страницы------------------------------